# PREMIUM: LLM PERSONALIZATION WITH INDIVIDUAL-LEVEL PREFERENCE FEEDBACK

## ABSTRACT

With an increasing demand for LLM personalization, various methods have been developed to deliver customized LLM experiences, including in-context learning, retrieval augmentation, and parameter-efficient fine-tuning. However, most existing methods are not readily locally deployable, limited by the compute cost, privacy risks, and an inability to adapt to dynamic user preferences. Here, we propose to use a tag system to efficiently characterize user profiles, inspired from the insights from personality typology and recommendation systems. Based on the observation, we present a locally deployable LLM-agnostic framework for achieving LLM personalization: **PREMIUM** (**P**reference **R**anking **EM**powered **I**ndividual **U**ser **M**odeling), which obtains individual-level feedback by having users rank responses and continuously self-iterates optimization during the interaction between the user and the LLM. Notably, a variant of PREMIUM, PREMIUM-Embed, can effectively capture user preferences while being deployable with laptop-level resources. Besides algorithmic innovation, we further prepare a novel dataset, Ranking-TAGER, which provides a valuable evaluation protocol for LLM personalization. Extensive experiments validate that PREMIUM remarkably outperforms various baselines, achieving a 15%-50% higher accuracy and a 2.5%-35% higher win rate on Ranking-TAGER, as well as a 3%-13% higher accuracy and a 2%-7.5% higher F1 Score on LaMP-2. More importantly, we further demonstrate that PREMIUM can develop an effective strategy with minimal interactive data, adapt to dynamic user preferences, and demonstrate excellent scalability in both scale and functionality.

## 1 INTRODUCTION

LLM-powered conversational agents have become increasingly prevalent (Jörke et al., 2024; Abbasian et al., 2024; Bagdasaryan et al., 2024), attracting a growing user base and amplifying the importance of LLM personalization. Personalized LLMs can be applied to a wide range of downstream tasks, encompassing customer service (Rome et al., 2024), personal health (Abbasian et al., 2024), recommender systems (Li et al., 2024), making it a valuable area of research.

To achieve alignment between LLMs and user preferences, existing research mainly falls into three categories: parameter-efficient fine-tuning (PEFT), retrieval-augmented LLMs (RALM), and in-context learning (ICL). (1) *PEFT-Based methods*, such as Baize (Xu et al., 2023), utilize user information to fine-tune open-source LLMs for generating user-preferred responses (Zhang et al., 2024b). However, such approaches are not applicable to black-box LLMs with proprietary parameters (such as GPT-3.5 and Gemini), greatly limiting their applicability, and fine-tuning LLMs imposes a burdensome cost on users. (2) *RALM-Based methods*, such as OPPU (Tan et al., 2024), incorporate retrieved user personal information into prompts to generate responses aligned with user preferences (Salemi et al., 2024b; Du et al., 2024). However, retrieval-augmented methods require users to provide a large amount of textual personal information, which may be challenging and pose potential privacy risk (Kirk et al., 2024). (3) *ICL-Based methods*, such as TidyBot (Wu et al., 2023), set explicit textual user profiles for users (Zhang et al., 2018) and leverage these user profiles through in-context learning (Dong et al., 2023) to achieve LLM personalization. While this approach offers advantages such as simplicity, the user information it requires raises potential privacy concerns (Kirk et al., 2024). Additionally, fixed user profiles cannot adapt to changes in user preferences (Kangaslahti & Alvarez-Melis, 2024; Shi et al., 2024) or provide query-related contexts to LLMs. Overall, existing

Table 1: **PREMIUM is an *LLM-agnostic* framework that does not require users to provide *personal textual information* and can *adapt to dynamic user preferences*, assisting LLMs in achieving *query-related* personalization.** Here is the comparison of PREMIUM and existing LLM personalization methods.

| Method | LLM-Agnostic | Textual Info. Free | Dynamic-Preference-Adaptive | Query-Related |
|---|---|---|---|---|
| Baize (Xu et al., 2023) | ✗ | ✗ | ✗ | ✓ |
| OPPU (Tan et al., 2024) | ✓ | ✗ | ✗ | ✓ |
| TidyBot (Wu et al., 2023) | ✓ | ✗ | ✗ | ✗ |
| PREMIUM (Ours) | ✓ | ✓ | ✓ | ✓ |

methods for LLM personalization still exhibit fundamental limitations in terms of flexibility, privacy security, and cost efficiency.

Psychological theories about personality typology reveal that individuals can be categorized into different personality types by assigning them "words that represent their preferences." (Myers, 1985; Keirsey, 1998). This method of characterizing individual personality is similar to tag-based approaches in recommendation systems (Belém et al., 2017; Furtado & Esmin, 2023). Inspired by these theoretical insights and practical experiences, we introduce a more rational and efficient method for characterizing user profiles - the Tagging System, which models user profiles by assigning tags that represent their personality traits and preferences.

Building on this foundation, we propose **PREMIUM** (**P**reference **R**anking **EM**powered **I**ndividual **U**ser **M**odeling), a novel LLM-agnostic framework for LLM personalization. Our key insight is that by having users rank responses based on their personal preferences, we obtain individual-level feedback, and leverage this feedback to continuously self-iterate optimization during the interaction between the user and the LLM, thereby aligning with the user's personal preferences. Furthermore, we implement two variants of PREMIUM: PREMIUM-Prompt and PREMIUM-Embed. The comparison between PREMIUM and some representative existing methods can be found in Table 1.

Besides algorithmic innovation, we further propose a novel dataset, Ranking-TAGER, which provides a valuable evaluation protocol for LLM personalization. Our comprehensive experiments on Ranking-TAGER validate that PREMIUM remarkably outperforms all baselines by achieving a 15%-50% higher accuracy and a 2.5%-35% higher win rate. Moreover, we further demonstrate some exciting findings: PREMIUM can develop an effective strategy with minimal interactive data, adapt to dynamic user preferences, and demonstrate excellent scalability in both scale and functionality.

In summary, our main contributions are as follows: (1) PREMIUM, a novel LLM-agnostic framework for achieving LLM personalization, to our knowledge, the first method that utilizes tags to characterize user profiles and leverage individual-level preference feedback to achieve LLM alignment with user preferences. (2) PREMIUM can be deployable locally with laptop-level resources, and consistently outperforms all baselines, achieving a 15%-50% higher accuracy and a 2.5%-35% higher win rate on Ranking-TAGER, as well as a 3%-13% higher accuracy and a 2%-7.5% higher F1 Score on LaMP-2. (3) Ranking-TAGER, a novel dataset that collects diverse user preferences and contributes to research on LLM Personalization, recommendation systems, and psychology studies.

## 2 PREMIUM: A Novel LLM-agnostic Personalization Framework

**Framework Overview** Fig. 1 offers an overview of the proposed PREMIUM framework. Our key insight is that by selecting tags to guide the LLM in generating responses with corresponding domain-specific elements, and by collecting user preference rankings for multiple responses, PREMIUM can utilize this individual-level feedback to continuously self-iterate optimization during the user-LLM interaction process, ultimately enabling the LLM to generate user-preferred responses.

**Responses Generation through the Tagging System** One key aspect of LLM personalization lies in the characterization of user profiles. To explore a more reasonable way of characterizing individual preferences, we draw upon theoretical support from psychological theories: In personality typology, some theories categorize individuals into different personality types by assigning them "words that represent their preferences." (Myers, 1985; Roccas et al., 2002) This method of characterizing personality is similar to tag-based approaches in recommendation systems (Belém et al., 2017; Furtado & Esmin, 2023), which are widely used for their efficiency and simplicity. In this work, we adopt a similar approach and propose a tag-based user profiling method - the Tagging System:

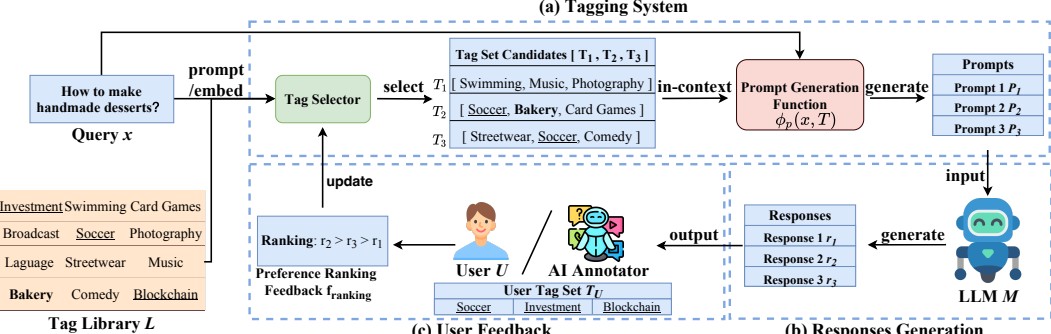

Figure 1: **PREMIUM Framework.** **(a) Tagging System:** Given a query $q$, the Tag Selector selects multiple Tag Set Candidates from the Tag Library, which are then transformed with $q$ into prompts by a Prompt Generation Function. **(b) Responses Generation:** Given prompt $P$, LLM $M$ generates response $r$. **(c) User Feedback:** Given multiple responses, the user (or AI Annotator) provides Preference Ranking Feedback, which is used to update the Tag Selector for the next interaction.

**Definition 1 (Tag Library).** To characterize users' preferences, we construct a Tag Library $L = \{t_1, t_2, \ldots, t_n\}$, consisting of $n$ tags representing domains of potential interests to users ("Investment", "Baking", "Biology", etc.). These tags cover 20 areas such as "Finance", "Athletics", and "Education".

**Definition 2 (User Tag Set).** For a specific user $U$, we assume they are interested in $k$ domains represented by the tags from the Tag Library, We use the tag set composed of the corresponding $k$ user tags as the user profile, refer to it as User Tag Set $T_U = [t_{U_1}, t_{U_2}, ..., t_{U_k}]$.

**Definition 3 (Tag Set Candidate).** For a query $q$ provided by user $U$, we select $k$ tags from the Tag Library $L$ to form a tag set $T = [t_{n_1}, t_{n_2}, ..., t_{n_k}]$, refer to it as a Tag Set Candidate.

**Definition 4 (Tag Selector).** Given a query $q$ provided by user $U$, the Tag Selector selects $m$ sets of Tag Set Candidates $[T_1, T_2, ...T_m]$ from the Tag Library $L$. These $m$ Tag Set Candidates assist LLM $M$ in generating $m$ distinct responses $[r_1, r_2, ...r_m]$ for $q$.

When the Tag Selector selects Tag Set Candidates from the Tag Library, we construct a prompt for the LLM by combining each Tag Set Candidate with the user query. We employ prompt engineering techniques to ensure that the LLM incorporates elements, perspectives, examples, and terminologies related to the tags present in the candidate into its generated response.

Specifically, by using a prompt generation function $\phi_p$, we transform each Tag Set Candidate $T_i$ and query $q$ into a prompt $P_i$: $P_i = \phi_p(q, T_i), i \in \{1, 2, \ldots, m\}$. By feeding the prompt $P_i$ into the LLM $M$, we obtain a response $r_i$ that is relevant to the tags in $T_i$: $r_i = M(P_i), i \in \{1, 2, \ldots, m\}$. The prompt template used to combine the user query and the candidate tag set is presented in Figure 6 of Appendix B.

**Objective of Responses Generation** Our aim is to make the responses (1) relevant to the domains the user is interested in, (2) adhere to the user's instructions and answer the user's questions. The former goal requires the selected tags to be within User Tag Set $T_U$, while the latter goal may require the selected tags to be relevant to the query $q$. For example, if a user interested in "Nutrition" asks, *"How to make handmade desserts?"*, our goal is to select the tags "Nutrition" and "Bakery" from the tag library to assist the LLM in generating a response such as *"To make handmade desserts with a focus on nutrition, consider using whole grain flour, natural sweeteners, and healthy fats...."*

**Preference Ranking Feedback on Responses** This paper focuses on utilizing individual-level feedback to facilitate LLM personalization. In this work, we adopt Preference Ranking Feedback $f_{ranking}$ as a signal for uncovering user preferences. Specifically, for each query $q$, the user $U$ is provided with $m$ responses $[r_1, r_2, ..., r_m]$. User $U$ is required to provide a preference ranking $f_{ranking}$ for these $m$ responses as individual-level preference feedback, which is used to update the Tag Selector for the next interaction. Thus, PREMIUM continuously self-iterates optimization during the interaction between the user and the LLM to assist LLM in generating user-preferred responses.

Here, we define the **Interaction History** of a complete interaction between the user and the LLM:

**Definition 5 (Interaction History).** An Interaction History $h$ consists of the following components: the user query $q$, $m$ Tag Set Candidates $[T_1, \ldots, T_m]$, their corresponding generated responses $[r_1, \ldots, r_m]$, and the user's Preference Ranking Feedback $f_{ranking}$.

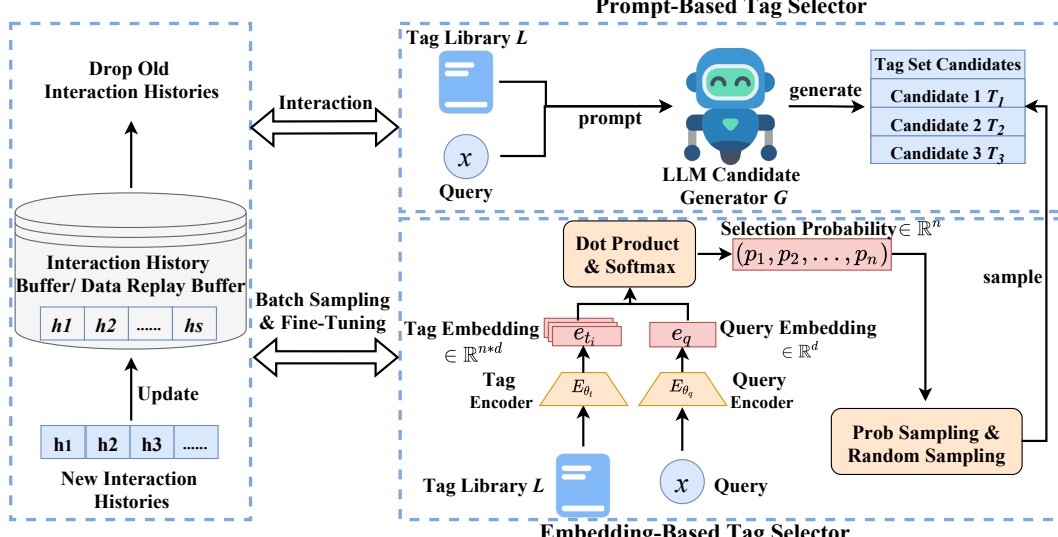

Figure 2: **Prompt-Based and Embedding-Based Tag Selector. Prompt-Based:** As shown in the upper box, given the query and Interaction Histories in the Buffer, LLM Candidate Generator selects Tag Set Candidates from Tag Library. **Embedding-Based:** As shown in the lower box, given the query, Tag Encoder and Query Encoder calculate the selection probability. Tag Set Candidates are selected through probability sampling and random sampling. After several interactions, data is sampled from Data Replay Buffer to update the Encoders.

Notably, PREMIUM is a concise and effective framework for achieving LLM personalization, without requiring access to any parameters, representations, or token probabilities of the LLMs. This makes it suitable for both parameter-open LLMs (such as LLaMA-2) and black-box LLMs (such as GPT-3.5).

# 3 PREMIUM-PROMPT: A SIMPLIFIED PROOF-OF-CONCEPT

First, we propose a relatively intuitive Prompt-Based implementation of PREMIUM. Prompt-Based methods have gained widespread adoption in many works due to its simplicity and the excellent reasoning capabilities of LLMs (Wu et al., 2023; Zeng et al., 2022; Zhang et al., 2024a).

**Prompt-Based Tag Selector** In this approach, we construct the Tag Selector in the PREMIUM by introducing an additional **LLM Candidate Generator** $G$ to infer the user's domains of interest from the Interaction Histories. Then, by combining the user's interests with a new query $q$ provided by them, the LLM Candidate Generator $G$ selects Tag Set Candidates from the Tag Library $L$. Figure 2 offers an overview of the Prompt-Based Tag Selector.

Specifically, we maintain an **Interaction History Buffer**, which stores the most recent $s$ Interaction Histories. Given a new query $q$, we submit the Interaction Histories from the Buffer along with $q$ to the LLM Candidate Generator $G$, prompting it to generate a Tag Set Candidate: $T = G(q, [h_1, \ldots, h_s])$.

Through repeating this process $m$ times, we obtain $m$ Tag Set Candidates, which are used to generate $m$ responses for User $U$ to rank. When User $U$ provides Preference Ranking Feedback $r_{ranking}$, we obtain the Interaction History $h$ for this interaction and update the Interaction History Buffer.

**Limitations in Real Applications** While Prompt-Based approaches benefit from simplicity, some works applying them to combinatorial optimization problems have shown drawbacks such as instability and degradation in performance as action space increasesYang et al. (2024)Liu et al. (2024).

We conduct experiments to validate the effectiveness of PREMIUM-Prompt in real applications. The experimental results and detailed analysis can be found in Appendix D.

When the action space is relatively small, PREMIUM-Prompt, while being concise, manages to uncover a portion of user tags, demonstrating good effectiveness and indirectly validating the rationality of our framework. However, when the action space is relatively large, it fails to model user preferences effectively, which may be attributed to its limited exploration capability. Furthermore, the buffer size is limited by LLM's effective context length, which severely restricts the LLM Candidate Generator's ability to learn user preferences from Interaction Histories. Additionally, PREMIUM-Prompt also suffers from high usage cost, unstable performance and sensitivity to prompts.

# 4 PREMIUM-EMBED: EFFECTIVE AND LIGHTWEIGHT IMPLEMENTATION

To address the various issues of PREMIUM-Prompt, we propose PREMIUM-Embed. In this variant, we encode the user preferences learned during the interaction process into the parameters of neural networks, thus overcoming the limitations of capacity and stability inherent in PREMIUM-Prompt, which are constrained by the limited size of the Interaction History Buffer.

**Embedding-Based Tag Selector** In this approach, we construct the Tag Selector by introducing two encoders: Query Encoder $E_{\theta_q}$ and Tag Encoder $E_{\theta_t}$, to respectively encode the semantic information of queries and tags. We then perform fine-tuning through $f_{ranking}$ to incorporate user's personal preferences into their parameters. Figure 2 offers an overview of Embedding-Based Tag Selector.

Specifically, given a user query $q$ and a tag $t_i$ from the Tag Library, we utilize the Query Encoder $E_{\theta_q}$ and the Tag Encoder $E_{\theta_t}$ to obtain the query embedding $e_q$ and tag embedding $e_{t_i}$ respectively (Lin et al., 2023; Lee et al., 2019): $e_q = E_{\theta_q}(q) \in \mathbb{R}^d, e_{t_i} = E_{\theta_t}(t_i) \in \mathbb{R}^d$.

We encode the semantic information of query $q$, tag $t_i$, and the preferences of user $U$ into two vectors of equal dimensions, $e_q$ and $e_{t_i}$. Leveraging these two heterogeneous embeddings, we can compute the probability of selecting $t_i$ into Tag Set Candidate. Here, we calculate the dot product of the tag embedding $e_t$ with the query embedding $e_q$ for each tag in the Tag Library, and apply the Softmax function to these scalars to compute the probability $p$ of selecting each tag: $p_i = \frac{e^{e_p \cdot e_{t_i}}}{\sum_{j=1}^n e^{e_p \cdot e_{t_j}}}$.
Utilizing the probability $p$, we select Tag Set Candidates from the Tag Library $L$.

**Tag Selector Training through Pairwise Preference Data** To learn the preferences of user $U$ from the Preference Ranking Feedback $f_{ranking}$, we decompose $f_{ranking}$ into **Pairwise Preference Data** $\{(w_i, l_i)\}_{i=1}^N$, where $w_i$ precedes $l_i$ in $f_{ranking}$. When the user provides a ranking of $m$ responses, we can obtain $N = \frac{m \times (m-1)}{2}$ pairs of Pairwise Preference Data, applying ranking as the form of feedback enables us to enhance data collection efficiency.

**Preference Loss Function** In $\{(w_i, l_i)\}_{i=1}^N$, the $w_i$-th response is preferred over the $l_i$-th, indicating that for a given query $q$, the tags generating the $w_i$-th response should be selected with a higher probability than those generating the $l_i$-th. We design the Preference Loss Function as follows:

$$L_p(\theta_q, \theta_t) = -\frac{1}{N} \sum_{i=1}^N \log(\sigma(\sum_{j=1}^k (E_{\theta_q}(q) \cdot E_{\theta_t}(t_j^{w_i})) - \sum_{j=1}^k (E_{\theta_q}(q) \cdot E_{\theta_t}(t_j^{l_i})))). \quad (1)$$

where $t_j^{w_i(l_i)}$ denotes the $j$-th tag in the $w_i(l_i)$-th Tag Set Candidate, $k$ is the number of tags in each Tag Set Candidate, and $\sigma$ represents the sigmoid function.

Our Preference Loss Function is similar in form to the RM loss proposed by Stiennon et al. (2022), which has been widely used in the reward modeling phase of the RLHF algorithms, demonstrating a strong capability to align with human preferences (Ouyang et al., 2022).

**Trade-off between Exploration & Exploitation** Uncovering user preferences involves two aspects: 1) "Exploration" of new user tags, which requires selecting tags that haven't yet been chosen to enter the Tag Set Candidates to obtain feedback. 2) "Exploitation" of current potential user tags, which requires selecting tags that have received some positive feedback. However, the number of tags selected at each interaction is limited, leading to a conflict between exploration & exploitation.

To achieve a trade-off between exploration & exploitation, we apply the following techniques during training: When selecting Tag Set Candidates, some candidates are chosen by the Embedding-Based Tag Selector to encourage exploitation, while others are randomly selected to encourage exploration.

Additionally, to enhance the model's ability for exploration in large action spaces, we introduce an auxiliary Entropy Loss Function $L_e$:

$$L_e = \sum_{i=1}^n p_i \log(p_i). \quad (2)$$

where n is the Tag Library size, $p_i$ represents the probability of selecting the $i$-th tag.

The $L_e$ term, by incorporating the negative entropy of the probability distribution of selected tags into the training loss, helps prevent the Embedding-Based Tag Selector from being confined to a subset of the Tag Library, thereby enhancing its exploration capability.

Finally, we integrate $L_p$ with $L_e$ weighted by a scalar $\delta$ into our loss function: $L = L_p + \delta L_e$.

**Enhancing training stability and data utilization** To enhance training stability and data utilization, we maintain a Data Replay Buffer with a maximum length $s$ and a first-in-first-out mechanism during training. After a certain number of interactions between the user and the LLM, we use the Interaction Histories to update the Data Replay Buffer. We then sample batches of data from the buffer to update the Tag Selector's parameters, using the updated Selector to assist in the next round of interactions.

## 5 RANKING-TAGER: DATASET FOR TAG-ASSISTED LLM PERSONALIZATION

Existing datasets designed for LLM personalization are well-constructed but predominantly incorporate textual user profiles (Salemi et al., 2024b; Du et al., 2024; Aliannejadi et al., 2024). This approach, however, carries potential privacy risks in practical applications (Kirk et al., 2024). To address these issues, we introduce an innovative dataset, **Ranking-TAGER** (**Ranking** - **T**ag-**A**ssisted **GE**nerated **R**esponses). Our dataset comprises 79,017 data entries and we partition it into three parts based on task categories: **Ranking-TAGER-RW (Routine Writing), Ranking-TAGER-SG (Story Generation)**, and **Ranking-TAGER-IF (Instruction Following)**. An overview of them, the detailed dataset format, query source and details of the collection process can be found in Appendix E.

**AI Annotator** In this work, we utilize LLM automatic annotation, which has seen widespread adoption in recent research involving human feedback (Bai et al., 2022; Dubois et al., 2024; Lee et al., 2023). Specifically, we employ an AI Annotator to provide Preference Ranking Feedback. It will rank responses based on how well they adhere to the user's instructions and how relevant they are to the user's domains of interest. Here, we choose "Qwen1.5-72B-Chat" as our AI Annotator due to its strong alignment with human capabilities (Bai et al., 2023).

**Benefits and Contributions** Ranking-TAGER offers several advantages over existing datasets: (1) It employs the Tagging System to characterize user profiles, which is a more realistic, reasonable, and concise approach. Moreover, it does not include text information from individual users, thus eliminating the risk of information leakage (Kirk et al., 2024). (2) Our dataset collects diverse preferences from 862 different users, which is difficult to obtain in reality. Additionally, when facing real users, the preferences collected in our dataset may fully or partially reflect their personal preferences. Therefore, leveraging our dataset can help LLMs quickly adapt to real user preferences (Kang et al., 2024). (3) The AI feedback included in Ranking-TAGER contains the Explanation conducted by the AI annotator before providing Preference Ranking. This makes our data highly interpretable and supports deeper analysis, which can be used in various fields such as LLM personalization, recommendation systems, and psychology studies.

## 6 EXPERIMENT

**Baselines.** To gain a comprehensive understanding of our PREMIUM's performance in assisting LLMs in generating user-preferred responses, we have adopted several baselines. All experiments are conducted under the same LLM: Mistral-7B (Jiang et al., 2023). Note that for all methods requiring training of neural networks, we initialize parameters using DRAGON-RoBERTa (Lin et al., 2023).

(1) **Vanilla LLM**: To examine the enhancement in LLMs' capability for user personalization brought about by PREMIUM, we compare it with the vanilla LLM with randomly selected Tag Set Candidates; (2) **RALM**: To investigate the enhancement in LLMs' personalization capability achieved through learning user preferences via Preference Ranking Feedback, we establish a baseline using the initial Tag Selector without fine-tuning. Specifically, we utilize a deep learning-based retriever, DRAGON (Lin et al., 2023), for selecting Tag Set Candidates; (3) **Population-Based Alignment**: To compare the performance of PREMIUM with existing alignment approaches that align LLMs with diverse human preferences on LLM personalization, we utilize feedback from 10 users with diverse preferences and employ our method for training; (4) **TidyBot**: We use TidyBot (Wu et al., 2023), a representative ICL-based method, as a baseline. It utilizes LLMs to summarize user profiles from interaction histories for personalization. TidyBot focuses on the personalized room organization task, and we

have attempted to adapt this method for the task of personalized response generation; (5) **OPPU**: We reproduce OPPU (Tan et al., 2024), a novel RALM-based method, based on the descriptions provided in their paper as one of our baselines. It incorporates user profile text along with retrieved personal information into prompts to generate personalized responses.

Notably, TidyBot and OPPU do not rely on our proposed tag system. To facilitate a comparison with these methods, we utilize the queries and the most preferred responses from the user-LLM interaction process of PREMIUM to form the user history, which serves as the textual user information relied upon by OPPU and TidyBot. This approach enables the baselines to benefit from user-selected data through ranking feedback, thereby enhancing their personalization capabilities.

**Additional Dataset.** To conduct a more comprehensive evaluation of our proposed PREMIUM framework, we utilize LaMP (Salemi et al., 2024b) as an additional dataset. Our work is based on LaMP-2 (Personalized Movie Tagging) from LaMP, where we incorporate personalized preference feedback to enable a comparison of PREMIUM-Embed with OPPU and TidyBot. Specifically, we employed the predefined movie tag pool from LaMP-2 as the tag library and provided ranking feedback for multiple responses based on the ground truth user responses available in LaMP-2.

**Metrics.** Our evaluation approach encompasses both automated and AI-based assessments: For Ranking-TAGER, we use the following two metrics: (1) **Accuracy**: This metric computes the proportion of tags selected to enter the Tag Set Candidates that are present in the User Tag Set. The closer it is to 1, the deeper the system's grasp of user preferences. (2) **Win Rate**: Besides Accuracy, we incorporate feedback from the AI annotator as another metric. The percentage represents the frequency of a response being chosen over our PREMIUM-Embed. A rate below 50% suggests that PREMIUM-Embed is outperforming the compared baseline. Compared to Accuracy, this provides a more comprehensive assessment: In addition to the selection of user tags, it considers other factors influencing user preferences, such as improved response quality from selecting query-relevant tags; For LaMP-2, we follow Salemi et al. (2024b) and utilize **Accuracy** and **F1 Score** as our metrics. Higher accuracy and F1 scores indicate more precise predictions for personalized movie tagging.

**Setups.** For the baselines based on the tag system, we use Ranking-TAGER as the dataset. To demonstrate the effectiveness of PREMIUM with different tag systems, we conduct experiments under three setups with increasing action spaces: "3/20," "3/50," and "3/100." The first number represents the number of tags in both the User Tag Set and the Tag Set Candidate, while the second number indicates the size of the Tag Library. Notably, under the same setup, all methods requiring user feedback use the same number of cases: 67, 112, and 164 for the three setups, respectively. In each of our experiments, the User Tag Set is randomly selected from the Tag Library to initialize the user's preferences. For TidyBot and OPPU, as stronger baselines, we compare them with PREMIUM-Embed across both the Ranking-TAGER and LaMP-2 datasets, providing a more comprehensive and convincing evaluation.

## 6.1 RESULTS ON PERSONALIZED USER PREFERENCE MODELLING EVALUATION

**PREMIUM-Embed achieves best performance among all datasets and all setups.** We report the performance of our methods and baselines in Table 2 and Table 3. (1) Across all datasets, PREMIUM-Embed significantly outperforms all baselines: for Ranking-TAGER, it achieves a 15%-50% accuracy advantage and a 2.5%-35% win rate advantage; for LaMP-2, it achieves a 3%-13% accuracy advantage and a 2%-7.5% F1 Score advantage. This suggests that using a tag system and leveraging individual-level preference feedback can effectively explore user preferences and assist LLMs in generating responses that align with them. (2) For baselines not based on preference feedback, vanilla-LLM and RALM fail to achieve satisfactory accuracy, underscoring the importance of preference feedback in modeling user preferences. However, RALM, benefiting from its retrieval capability to select tags relevant to queries, manages to achieve a suboptimal win rate in some setups. (3) Population-Based Alignment utilizes preference feedback, allowing it to achieve suboptimal accuracy and win rates in certain setups. However, due to inconsistencies in the feedback it aligns with, it fails to match the performance of PREMIUM-Embed. This highlights the challenges faced by methods that align diverse population preferences when assisting LLMs in generating responses preferred by individual users. (4) PREMIUM-Prompt exhibits unstable accuracy but consistently high win rates in small action spaces, indicating a stronger capability of the LLM Candidate Generator to select tags relevant to user queries compared to exploring user preferences during interactions. (5)

Table 2: **PREMIUM-Embed consistently outperforms tag-system-based baselines among all setups.** Bold and underline denote the best and second-best results. All results are obtained by averaging the results of multiple experiments. PREMIUM-Prompt is only included in the "3/20" setup comparison due to its relatively poor performance in large action spaces.

| Dataset | Ranking-TAGER-RW | | | | | |
|---|---|---|---|---|---|---|
| Setup | 3/20 (67 Cases) | | 3/50 (112 Cases) | | 3/100 (164 Cases) | |
| Method | Accuracy | Win Rate | Accuracy | Win Rate | Accuracy | Win Rate |
| **Vanilla LLM** | 15.00% | 14.17% | 6.00% | 15.00% | 3.00% | 17.71% |
| **RALM** | 16.04% | 18.33% | 8.33% | 23.33% | 1.65% | 29.57% |
| **Population-Based Alignment** | 29.44% | 13.33% | 22.25% | 20.00% | 11.00% | 25.30% |
| **PREMIUM-Prompt (Ours)** | 6.11% | 35.00% | / | / | / | / |
| **PREMIUM-Embed (Ours)** | **54.32%** | **50.00%** | **55.77%** | **50.00%** | **35.23%** | **50.00%** |
| Dataset | Raning-TAGER-SG | | | | | |
| Setup | 3/20 (67 Cases) | | 3/50 (112 Cases) | | 3/100 (164 Cases) | |
| Method | Accuracy | Win Rate | Accuracy | Win Rate | Accuracy | Win Rate |
| **Vanilla LLM** | 15.00% | 14.17% | 6.00% | 16.67% | 3.00% | 13.50% |
| **RALM** | 10.59% | 12.50% | 3.12% | 16.67% | 2.05% | 21.67% |
| **Population-Based Alignment** | 22.07% | 25.00% | 14.75% | 30.00% | 8.56% | 23.33% |
| **PREMIUM-Prompt (Ours)** | 28.61% | 36.67% | / | / | / | / |
| **PREMIUM-Embed (Ours)** | **60.74%** | **50.00%** | **46.90%** | **50.00%** | **23.25%** | **50.00%** |
| Dataset | Ranking-TAGER-IF | | | | | |
| Setup | 3/20 (67 Cases) | | 3/50 (112 Cases) | | 3/100 (164 Cases) | |
| Method | Accuracy | Win Rate | Accuracy | Win Rate | Accuracy | Win Rate |
| **Vanilla LLM** | 15.00% | 28.89% | 6.00% | 30.83% | 3.00% | 25.19% |
| **RALM** | 19.25% | 39.87% | 6.95% | 33.33% | 2.39% | 31.09% |
| **Population-Based Alignment** | 33.62% | 35.66% | 14.86% | 32.50% | 4.87% | 33.49% |
| **PREMIUM-Prompt (Ours)** | 10.56% | 45.02% | / | / | / | / |
| **PREMIUM-Embed (Ours)** | **62.99%** | **50.00%** | **38.12%** | **50.00%** | **25.27%** | **50.00%** |

Table 3: **PREMIUM-Embed consistently outperforms OPPU and TidyBot across all datasets.** For Ranking-TAGER, we utilize only the "3/50" setup and Accuracy metric because TidyBot and OPPU depend on user interaction history and do not employ a tag system. For all methods, we do not use PEFT due to its high computational cost. For OPPU, we selected three different settings $k = 1, 2, 4$ as baselines, where $k$ represents the top-$k$ data retrieved and integrated into the prompt during the RAG process.

| Dataset | LaMP | | Ranking-TAGER | | |
|---|---|---|---|---|---|
| Subset | LaMP-2 | | RW | SG | IF |
| Method\Metric | Accuracy | F1 Score | Win Rate | Win Rate | Win Rate |
| **TidyBot** | 20.00% | 23.82% | 45.00% | 47.50% | 37.92% |
| **OPPU(k=1)** | 30.00% | 29.39% | 31.25% | 40.00% | 35.29% |
| **OPPU(k=2)** | 23.34% | 24.60% | 36.25% | 42.50% | 32.05% |
| **OPPU(k=4)** | 25.00% | 26.20% | 33.75% | 45.00% | 35.39% |
| **PREMIUM-Embed(Ours)** | **33.33%** | **31.46%** | **50.00%** | **50.00%** | **50.00%** |

For TidyBot and OPPU, despite feeding explicit user profiles and interaction histories to the LLM, they still do not achieve the same level of personalization as PREMIUM-Embed, demonstrating the limitations of LLMs in extracting diverse individual preferences from complex text, while also highlighting the advantages of PREMIUM over ICL-based and RALM-based methods.

**PREMIUM-Embed develops an effective strategy with minimal interactive data.** To validate that PREMIUM-Embed incurs a low "interaction cost," we trained our model using only 30 interaction data points in the "3/50" setup. After 30 interactions with the user, our method increased the average accuracy from 6.36% to 24.76%, achieving an average improvement of approximately 4 times. This suggests that our approach requires only a small amount of interaction data to rapidly adapt to a new user's preferences. Detailed experimental results can be found in Appendix G.1.

**Laptop-Level Resources Are Sufficient** The size of the model used in the Embedding-Based Tag Selector is within 1GB, making it lightweight and deployable locally. We trained our method on a Yoga Pro 14s ARH7 laptop, utilizing only CPU resources (8 cores, 3.20GHz frequency). We record the average training time and maximum memory consumption across three setups in Table 4.

## 6.2 NEW FINDINGS FROM OUR METHOD

**PREMIUM-Embed can make adaptation to dynamic user preferences.** In practical scenarios, the preferences of LLM users are not static but dynamically change over time (Kangaslahti & Alvarez-

Table 4: **Our method requires only laptop-level resources.**

| 3/20 Time Cost (s) | 3/50 Time Cost (s) | 3/100 Time Cost (s) | Max Memory Cost (MB) |
|---|---|---|---|
| 727.74 | 1728.66 | 3014.52 | 5937.54 |

Table 5: **Our method can simultaneously improve the accuracy of ordinary and binary tags.**

| Init Ord. Acc. | Trained Ord. Acc. | Init Bin. Acc. | Trained Bin. Acc. |
|---|---|---|---|
| 6.28% | 25.57% | 53.14% | 80.04% |

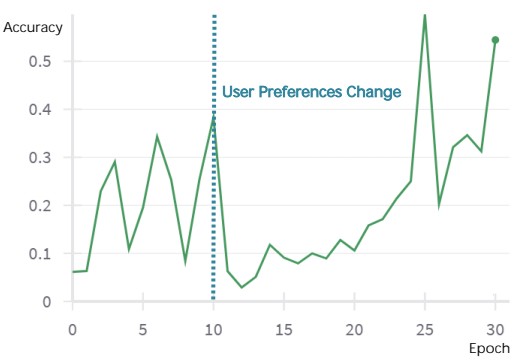

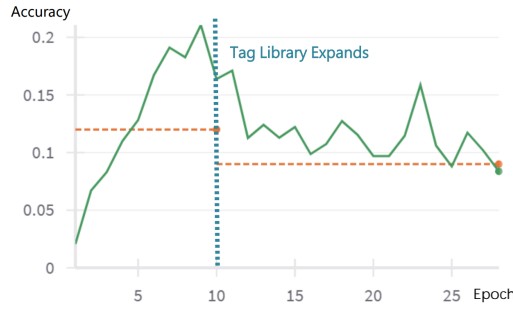

Figure 3: **PREMIUM-Embed can make adaptations to dynamic user preferences.** Within 50 interactions where user preferences changed, PREMIUM-Embed increases the accuracy beyond the accuracy before the user preferences changed.

Figure 4: **PREMIUM-Embed generalizes to expanded Tag Library.** The orange dashed line represents 6 times the accuracy of random selection. After the Tag Library expands, the accuracy of PREMIUM-Embed remains above the orange dashed line.

Melis, 2024; Shi et al., 2024), posing significant challenges for methods that apply fixed user profiles (Wu et al., 2023; Zhang et al., 2024b). To examine the effectiveness of our method in handling dynamic user preferences, we conduct the following experiments under the '3/50' setup: After 50 interactions between the user and LLM, we modify the user's preferences by changing two tags in the User Tag Set and then allow the user with the updated preferences to continue interacting with LLM. The experimental results, as shown in Figure 3, demonstrate that our method successfully adapts to new user preferences through new interaction data, illustrating the flexibility of our approach.

**PREMIUM-Embed can generalize to expanded Tag Library.** In real-world scenarios, as new popular interest domains emerge, there is a need to incorporate new tags into the Tag Library (Shi et al., 2024). Here, we validate that our method can generalize to an expanded Tag Library without retraining from scratch. We conduct the following experiments: Initially, the experimental setup is "2/100", and after fine-tuning for 10 epochs, we add 100 new tags to the Tag Library, including a new user tag. Therefore, we transform the setup to "3/200" and continue training. Figure 4 depicts our experimental findings, revealing that following the expansion of the Tag Library, PREMIUM-Embed effectively recognizes the new user tag during the interaction process. Furthermore, the multiplier of accuracy growth after expanding the Tag Library remains consistent with the pre-expansion multiplier when compared to random sample accuracy. This indicates that our method maintains its fundamental performance even as the Tag Library expands.

**PREMIUM-Embed can extend to binary tags.** When characterizing user profiles, some descriptions of preferences may be contradictory (Myers, 1985; Jang et al., 2023). In such cases, we need to use binary tags to model user preferences. Specifically, for each pair of binary tags, we choose one to represent the user's preference. To validate our method's extension to binary tags, we conduct the following experiments on the "3/50" setup: We augment the original Tag Library with four pairs of binary tags. During training, the Embedding-Based Tag Selector is responsible for selecting both types of tags simultaneously. The experimental results, as shown in Table 5, demonstrate that our method achieves synchronous improvements in accuracy on both ordinary tags and binary tags, confirming that our method can extend to binary tags.

**Ablation Study** We investigate the influence of different design choices on PREMIUM-Embed:
(1) **w/wo Data Replay Buffer**: In this variant, we remove the Data Replay Buffer, so each data point is only involved in one gradient computation. We leverage this to examine the impact of Data Replay

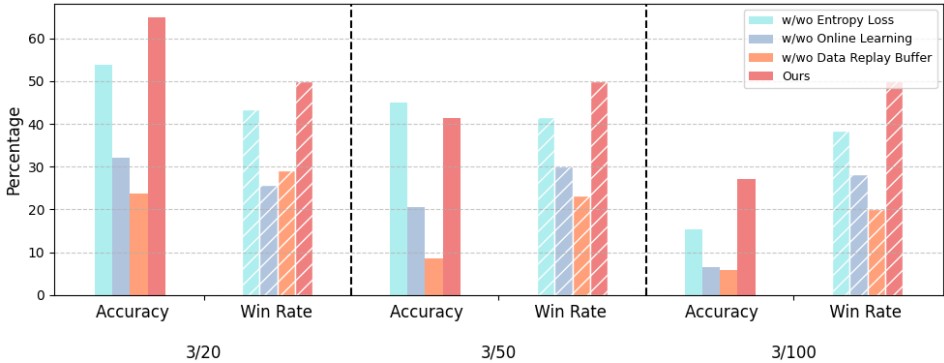

Figure 5: PREMIUM-Embed performs better than all the other variants on Ranking-TAGER-RW.

Buffer on our method in terms of higher training stability and data utilization efficiency.
(2) **w/wo Online Learning**: We explore the feasibility of an online learning setup in our approach, where the model interacts with the user to acquire new data and updates its parameters accordingly. In this variant, all the data we use is obtained from interactions between user and the initial model.
(3) **w/wo Entropy Loss**: In this variant, we remove the auxiliary Entropy Loss to evaluate its contribution to the trade-off between exploration & exploitation.

We report the evaluation results on Ranking-TAGER-RW in Fig. 5. It is clear from these comparisons that our method outperforms all the variants in most setups and metrics, demonstrating the validity of our training approach. We place the evaluation results on other datasets in Appendix G.3.

**Further Experiments** We also conduct experiments using LLMs of various sizes and architectures as the backbone, human evaluations, and experiments using only pairwise comparisons as user feedback. The details of these experiments are presented in Appendix H.

## 7 ADDITIONAL RELATED WORKS

**LLM Personalization** Recent research on LLM personalization has explored numerous directions: Collins et al. (2023) utilizes federated learning with PEFT to balance between personalization and robustness. Zhang et al. (2024c) employs a Bayesian Optimization searching strategy to find the optimal LoRA injection method in PEFT. Salemi et al. (2024a) attempts to use the discrepancy between responses generated by LLMs and ground truth responses in the dataset as a signal to fine-tune the retriever used for retrieval augmentation. Jin et al. (2024) applies the retrieval-augmented method to personal health management. Karra & Tulabandhula (2024); Yang et al. (2023); Liu et al. (2023); Chen et al. (2024) leverage the powerful summarization capabilities of LLMs to summarize user interaction histories, such as search and browsing records, into textual user profiles.

**Learning from Human Feedback** Learning from Human Feedback is widely employed to align LLMs with human values (Ziegler et al., 2020; Nakano et al., 2022). Reinforcement Learning from Human Feedback (RLHF) utilizes pairwise comparison feedback from annotators and RL to align LLMs with human values (Stiennon et al., 2022; Ouyang et al., 2022). Some works also attempt to extract human preferences by utilizing ranking feedback between responses (Yuan et al., 2023; Song et al., 2024). Additionally, some efforts involve directly fine-tuning LLMs using human feedback to address issues such as training instability in RL (Rafailov et al., 2023; Tang et al., 2024). However, most existing relevant works focus on aligning LLMs with diverse human values rather than the preferences of a specific individual user (Lanctot et al., 2023; Chakraborty et al., 2024). Applying Learning from Human Feedback methods to individual feedback remains an area yet to be explored.

## 8 CONCLUSION

In this study, we propose PREMIUM, an innovative LLM-agnostic framework for LLM personalization, which utilizes tags to characterize user profiles and individual-level preference feedback to align with user preferences, addressing the limitations of existing methods in flexibility, privacy, and cost. PREMIUM includes two variants: PREMIUM-Prompt and PREMIUM-Embed, with the latter excelling in performance and efficiency. Our Ranking-TAGER dataset, which provides a valuable evaluation protocol for LLM personalization, enabled us to demonstrate that PREMIUM outperforms all baselines by achieving significantly higher accuracy and win rates. Notably, PREMIUM-Embed requires minimal resources, can adapt to dynamic user preferences, generalize to expanded Tag Library, and extend to binary tags, making it a practical solution for personalized LLMs.

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

## A  Tag Library and Binary Tags

The tags in the **Tag Library** cover 20 different areas, allowing us to depict rich and diverse user profiles. The Tag Library used in this paper are shown as below:

1.  **Finance**: Investment, Banking, Accounting, Insurance, Stock market, Taxation, Retirement planning, Personal finance, Corporate finance, Venture capital

2.  **Athletics**: Running, Gymnastics, Swimming, Cycling, Martial arts, Yoga, CrossFit, Team sports, Extreme sports, Weightlifting

3.  **Gaming**: Role-playing games, Strategy games, Puzzle games, Simulation games, Action games, Adventure games, Casual games, Multiplayer games, Board games, Card games

4.  **Media**: Journalism, Broadcasting, Advertising, Social media, Public relations, Film production, Photography, Graphic design, Content creation, Podcasting

5.  **Health**: Nutrition, Exercise physiology, Mental health, Public health, Alternative medicine, Physical therapy, Chronic illness management, Aging and geriatrics, Epidemiology, Healthcare administration

6.  **Environment**: Conservation, Renewable energy, Pollution control, Sustainable agriculture, Wildlife preservation, Climate change mitigation, Environmental policy, Ecotourism, Environmental education, Green technology

7.  **Education**: K-12 education, Higher education, Online learning, Special education, Adult education, Educational technology, Curriculum development, Educational psychology, Vocational training, Language learning

8.  **Fashion**: Apparel design, Fashion photography, Fashion modeling, Textile design, Fashion merchandising, Sustainable fashion, Luxury fashion, Streetwear, Fashion blogging, Costume design

9.  **Travel**: Adventure travel, Cultural tourism, Ecotourism, Backpacking, Luxury travel, Solo travel, Family travel, Budget travel, Business travel, Food tourism

10. **Entertainment**: Music, Theater, Dance, Comedy, Magic, Circus, Cabaret, Variety shows, Performance art, Improvisation

11. **Technology**: Artificial intelligence, Internet of Things, Augmented reality, Virtual reality, Blockchain, Cybersecurity, Quantum computing, Biotechnology, Robotics, Nanotechnology

12. **Food**: Culinary arts, Baking, Pastry, Gastronomy, Food science, Nutrition science, Food safety, Organic farming, Food preservation, Fermentation

13. **Law**: Criminal law, Civil law, Constitutional law, Contract law, Family law, Corporate law, Intellectual property law, Environmental law, International law, Tax law

14. **Psychology**: Clinical psychology, Cognitive psychology, Developmental psychology, Social psychology, Educational psychology, Industrial-organizational psychology, Forensic psychology, Health psychology, Neuropsychology, Counseling psychology

15. **Science**: Physics, Chemistry, Biology, Astronomy, Geology, Environmental science, Neuroscience, Genetics, Meteorology, Ecology

16. **Art**: Painting, Sculpture, Drawing, Printmaking, Photography, Installation art, Performance art, Digital art, Mixed media, Street art

17. **Agriculture**: Crop science, Horticulture, Livestock farming, Aquaculture, Agribusiness, Sustainable agriculture, Precision agriculture, Agricultural engineering, Agricultural economics, Soil science

18. **Film**: Directing, Screenwriting, Cinematography, Film editing, Film production, Film criticism, Film theory, Documentary filmmaking, Animation, Independent film

19. **Pet**: Dog training, Cat care, Bird keeping, Aquarium keeping, Exotic pets, Pet grooming, Pet nutrition, Pet photography, Veterinary medicine, Pet adoption

20. **Policy**: Economic policy, Social policy, Environmental policy, Healthcare policy, Foreign policy, Education policy, Immigration policy, Fiscal policy, Criminal justice policy, Energy policy

Additionally, in Section 6.2 "PREMIUM-Embed can extend to binary tags," we design four pairs of **Binary Tags** to represent users' preferences for writing styles. The Binary Tags used in this paper are shown as below:

(Thorough, Brief); (Objective, Subjective); (Humorous, Serious); (Professional, Amateurish)

## B  PROMPT UTILIZATION

Here, we present the detailed prompt instructions used in our work:

(1) **Prompt Generation Function**: The prompts used for the Prompt Generation Function include two versions: the *"Ordinary Setup"* for regular experiments and the *"Binary Setup"* for Section 6.2 "PREMIUM-Embed can extend to binary tags.", as shown in Fig. 6.
(2) **AI Annotator**: The prompts used for the AI Annotator include two versions: the *"Ordinary Setup"* for regular experiments and the *"Binary Setup"* for Section 6.2 "PREMIUM-Embed can extend to binary tags.", as shown in Fig. 7 and 8, respectively.
(3) **LLM Candidate Generator**: The prompt used for LLM Candidate Generator is shown in Fig. 9 and 10.

**Prompt Generation Function:**

**Ordinary Setup:**

*System:*
"You are a helpful assistant. Please answer the user's question.
 Your answer should try to include relevant elements, perspectives,
examples, terminologies from the following daomains: {*Tag Set Candidate*}."

*User:*
{*query*}

**Binary Setup:**

*System:*
"You are a helpful assistant. Please answer the user's question.
Your answer should try to include relevant elements, perspectives,
examples, terminologies from the following domains: {*Tag Set Candidate*}.
Additionally, your answer should try to adhere to the following writing styles: {*Binary Tags*}."

*User:*
{*query*}

Figure 6: Prompt for Prompt Generation Function.

## C  INSIGHTS AND DISCUSSIONS ON THE DESIGN OF PREMIUM

### C.1  ADVANTAGES OF THE TAG SYSTEM

**The granularity of the tag system can cover sufficient diversity among humans**  Considering the "3/100" setup, its possible combinations can represent 160k different user types. Additionally, in Section 6.2, we discuss how our method can extend to binary tags. With 100 binary tags, the possible combinations can represent $2^{100}$ different user types, theoretically covering sufficient diversity among humans.

Notably, our method is scalable with the number of tags with linear compute and storage costs. In real-world applications, we could extend to thousands of tags to sufficiently achieve fine granularity.

## AI Annotator (Ordinary Setup):

You are an AI annotator responsible for ranking responses generated by LLM.
The User has interests in the following domains: {*user_tag_set*}!!!
Given the User Question and {*m*} responses generated by LLM, you need to rank the responses
based on how well they adhere to the User's instructions and answer the User's questions
and how relevant they are to the domains the User is interested in.

Before you rank the responses, you need to provide an Explanation for your judgment.
Please incorporate the User's interests into the Explanation!
Note: Responses may contain incorrect User's interests.
Please pay attention to identifying these errors and include them in the Explanation!
The actual User's interests are in the following domains: {*user_tag_set*}!!!

Ensure that the order of the responses does not influence your decision.
Do not let the length of the responses impact your evaluation.

The system's input is in this format:
[User Question]
{*query*}
[The Start of Response 1]
{*response_1*}
[The End of Response 1]
......
[The Start of Response {*m*}]
{*response_{m}*}
[The End of Response {*m*}]

Your answer must follow this format:
[Explanation]
{*Your Explanation*}
[Ranking]
{The ranking you provide. Use NUMBERs to represent responses, separated by ", ".
Do not include any characters other than Numbers, ",", and " "!!!
The number of NUMBERs appearing in the ranking must be consistent with
the number of responses! For example:{*ranking_example*}}
[The End of AI Feedback]

Figure 7: Prompt for AI Annotator in Ordinary Setup.

**Superiority of the Tag System to Alternative Solutions** Traditional LLM personalization methods, such as integrating user information into model parameters through PEFT, or utilizing textual user information with methods like RAG and ICL, exhibit fundamental limitations in terms of flexibility, privacy security, and cost efficiency.

While user tags are slightly less expressive, they offer significant advantages in privacy protection, cost, and efficiency compared to alternatives.

### C.2 ADVANTAGES OF PREFERENCE RANKING FEEDBACK

It is worth noting that the **Preference Ranking Feedback** we adopt has several advantages compared to the signals used in previous works:

• **Preference Ranking Feedback is both readily accessible and unbiased.** Unlike methods that necessitate users to provide "ground truth personalized responses" of their queries (Salemi et al., 2024b;a) or edit responses based on personal preferences (Gao et al., 2024), Preference Ranking

## AI Annotator (Binary Setup):

You are an AI annotator responsible for ranking responses generated by LLM.
The User has interests in the following domains: {*user_tag_set*}!!!
Additionally, the User prefers responses written in the following styles: {*binary_user_tag_set*}!!!
Given the User Question and {*m*} responses generated by LLM, you need to rank the responses
based on how well they adhere to the User's instruction and answer the User's question
and how relevant they are to the domains the User is interested in and
how closely they align with the writing styles preferred by the User.

Before you rank the responses, you need to provide an Explanation for your judgment.
Please incorporate the User's interests and preferences into the Explanation!
Note: Responses may contain incorrect User's interests and preferences.
Please pay attention to identifying these errors and include them in the Explanation!
The actual User's interests are in the following domains: {*user_tag_set*}!!!
The actual writing styles preferred by the User are: {*binary_user_tag_set*}!!!
***The remaining part is the same as Ordinary Setup ***

Figure 8: Prompt for AI Annotator in Binary Setup.

Feedback simply requires users to rank several responses to each query. This ranking task is easy to accomplish and results in much less bias. Moreover, the requirement for users to possess knowledge of the "ground truth" of their queries is inherently impractical (Salemi et al., 2024a).

• **Preference Ranking Feedback safeguards user privacy.** Some other methods require users to provide textual user information (Karra & Tulabandhula, 2024; Yang et al., 2023; Liu et al., 2023; Chen et al., 2024), which may introduce potential privacy risks (Kirk et al., 2024), whereas Preference Ranking Feedback does not require users to provide any textual data.

• **Preference Ranking Feedback is relevant to users' queries and can adapt to changes in user preferences.** Unlike some other methods that model fixed textual user profiles for users (Zhang et al., 2018), which cannot achieve query-related personalization and cannot accommodate changes in user preferences over time (Kangaslahti & Alvarez-Melis, 2024; Shi et al., 2024), Preference Ranking Feedback incorporates users' **real-time** preferences for responses **to specific queries**. This makes our approach query-related and able to adapt to changes in user preferences, as demonstrated in Section 6.2 "PREMIUM-Embed can make adaptation to dynamic user preferences."

• **Applying ranking as the form of feedback enables us to enhance data collection efficiency.** When the user provides a ranking of $m$ responses, we can obtain $N = \frac{m \times (m-1)}{2}$ pairs of Pairwise Preference Data.

### C.3 INSIGHT OF PREFERENCE LOSS

Here, we highlight the potential connection between our Preference Loss $L_p$ and a commonly used loss function in contrastive learning frameworks and representation learning, the InfoNCE Loss (Gutmann & Hyvärinen, 2010; Oord et al., 2018). The form of the Preference Loss is as follows:

$$L_p(\theta_q, \theta_t) = -\frac{1}{N} \sum_{i=1}^{N} \log \left( \sigma \left( \sum_{j=1}^{k} \left( E_{\theta_q}(q) \cdot E_{\theta_t}(t_j^{w_i}) \right) - \sum_{j=1}^{k} \left( E_{\theta_q}(q) \cdot E_{\theta_t}(t_j^{l_i}) \right) \right) \right).$$

where $\theta_q$ and $\theta_t$ denote the parameters of the Query Encoder $E_{\theta_q}$ and the Tag Encoder $E_{\theta_t}$, respectively. $t_j^{w_i(l_i)}$ represents the $j$-th tag in the $w_i(l_i)$-th Tag Set Candidate, $k$ is the number of tags in each candidate, and $\sigma$ represents the sigmoid function.

**LLM Candidate Generator (1):**

> You are an assistant tasked with building a User Profile for a specific User.
> The User has interests in certain specific domains.
> You will receive a Tag Library, a User Query, and a set of Interaction Histories for the User.
>
> Within each Interaction History, you will be provided with a Previous Query,
> {m} Tag Set Candidates, and the corresponding Responses
> of these Tag Set Candidates to the Previous Query.
> Additionally, the User's Preference Ranking for those {m} Responses will be provided.
> The Ranking is based on how well the Responses adhere to the User's previous instructions
> and answer the User's previous questions, as well as
> how relevant they are to the domains the User is interested in.
>
> Based on the Interaction Histories, you need to infer the User's potential domains of interest.
> You will then select "{k}" Tags from the Tag Library to form the User's Profile.
> These selected Tags should meet the following criteria:
> - They represent domains of interest to the User.
> - They are relevant to the content of the provided User Query.
> - They must be Tags that appear in the Tag Library provided!!!
>
> Note: the "Tag Set Candidate" in the [Interaction Histories] do not necessarily
> represent the domains that the User is actually interested in.
> They only represent the Tag Sets used to generate the corresponding responses.
> To determine the domains of actual interest to the User, you need to analyze the "User's
> Preference Ranking" provided in the [Interaction Histories] along with the "Tag Set Candidate".
>
> Before you provide a User Profile, you need to give an 'Explanation' that includes your analysis
> of the Interaction Histories, potential domains of interest for the User, and your reasons for
> selecting these {k} Tags as the User Profile.

Figure 9: Prompt for LLM Candidate Generator (1).

The form of the InfoNCE Loss is as follows:

$$\mathcal{L}_{\text{InfoNCE}} = -\sum_{i=1}^{N} \log \frac{\exp(\text{sim}(z_i, z_i^+)/\tau)}{\exp(\text{sim}(z_i, z_i^+)/\tau) + \sum_{j=1}^{K} \exp(\text{sim}(z_i, z_j^-)/\tau)}$$

where $\text{sim}(a, b)$ denotes a similarity function, often cosine similarity. $z_i$ and $z_i^+$ are the representations of a data point $x_i$ and its positive sample (e.g., an augmentation of $x_i$ ) $x_i^+$ respectively. $\tau$ is a temperature parameter that controls the sharpness of the distribution. $N$ is the batch size, and $K$ is the number of negative samples.

It aims to help the model learn representations by distinguishing between positive (related) and negative (unrelated) samples, maximizing mutual information between positive pairs while effectively discriminating against negative samples.

When $\text{sim}(a, b)$ is set to dot product and the temperature parameter $\tau$ is set to 1.0, the form of the Preference Loss aligns with that of the InfoNCE Loss: here, the query $q$ can be regarded as the data point $x_i$, $t_j^{w_i}$ can be regarded as the positive sample $x_i^+$, and $t_j^{l_i}$ can be regarded as the negative sample $x_i^-$.

This association can be understood as follows: In the InfoNCE Loss, labels are derived from the objective correlation between positive and negative samples, while the labels in our Preference Loss are based on the subjective preferences provided by the user. This indirectly explains why our method demonstrates a strong capability to align with user preferences and also showcases a potential new application scenario for the InfoNCE Loss.

## LLM Candidate Generator (2)

The system's input is in this format:
[Tag Library]
*{tag_library}*

[User Query]
*{user_query}*

[Interaction Histories]
[The Start of Interaction History 0]
Previous Query:
*{previous_query}*
Tag Set Candidate 1:
*{tag_set_candidate_1}*
Response 1:
*{response_1}*
......
Tag Set Candidate *{m}*:
*{tag_set_candidate_{m}}*
Response *{m}*:
*{response_{m}}*
User's Preference Ranking:
*{user_preference_ranking}*
[The End of Interaction History 0]
[The Start of Interaction History 1]
Previous Query:
*{previous_query}*
Tag Set Candidate 1:
*{tag_set_candidate_1}*
Response 1:
*{response_1}*
......
Tag Set Candidate *{m}*:
*{tag_set_candidate_{m}}*
Response *{m}*:
*{response_{m}}*
User's Preference Ranking:
*{user_preference_ranking}*
[The End of Interaction History 1]
......

Your answer must follow this
format:
[Explanation]
*{Your Explanation}*
[User Profile]
{'*{k}*' Tags from the Tag Library,
separated by ", "}
[The End of Answer]

Figure 10: Prompt for LLM Candidate Generator (2).

## D    EXPERIMENTS OF PREMIUM-PROMPT IN REAL APPLICATIONS

To validate the effectiveness of the PREMIUM-Prompt in real applications, we conducted experiments in two different setups: "3/20" and "3/50", here, the first number indicates the number of tags contained in the User Tag Set as well as the Tag Set Candidate, while the latter number represents the Tag Library size. In both setups, we set candidates num = 3, buffer size = 5, iteration num = 30.

We conducted 3 experiments in each setup and recorded the average accuracy of PREMIUM-Prompt on the test set after 30 iterations, as well as the average number of tokens used per iteration during the interaction with the LLM Candidate Generator. Here, the average number of tokens we recorded includes only the tokens present in the prompts submitted to the LLM Candidate Generator, the experimental results are shown in Table 6.

Table 6: Experimental results for PREMIUM-Prompt in Real Applications.

|  | 3/20 | | 3/50 | |
| --- | --- | --- | --- | --- |
|  | **Accuracy** | **AVG Tokens Num** | **Accuracy** | **AVG Tokens Num** |
| Experiment 1 | 0.33 | 7166 | 0.00 | 7191 |
| Experiment 2 | 0.03 | 7279 | 0.03 | 7268 |
| Experiment 3 | 0.67 | 7379 | 0.00 | 7304 |

When the action space is relatively small, such as in "3/20", the PREMIUM-Prompt, while being concise and easy to implement, manages to uncover a portion of user tags in a small number of interactions in over half of the experiments, demonstrating good effectiveness and indirectly validating the rationality of our framework.

However, when the action space is relatively large, as in "3/50", PREMIUM-Prompt fails to model user preferences effectively, which may be attributed to its limited exploration capability.

Furthermore, the average token consumption per interaction with the LLM Candidate Generator reveals that the buffer size $s$ is limited by the effective context length of the LLM. Considering that the effective context length of the LLM "Qwen1.5-72B-Chat" used in this experiment is 32K, the maximum buffer size is around 20. This severely restricts the LLM Candidate Generator's ability to learn user preferences from Interaction Histories.

## E    DETAILS OF RANKING-TAGER

### E.1    DATASET FORMAT

Each data entry in Ranking-TAGER includes the following components:

• "user tag set": $T_U$ of the user who annotates the data entry.

• "query": The query from the user.

• "Tag Set Candidates": Three Tag Set Candidates, each containing three tags.

• "Responses": Responses generated with the three Tag Set Candidates, generated by "Mistral-7B."

• "AI feedback": "AI feedback" consists of two parts: an Explanation for AI annotator's judgment and the Preference Ranking it provides (Wei et al., 2023).

• "pairwise preferences": Pairwise Preference Data derived from the Preference Ranking provided by AI annotator.

### E.2    QUERY SOURCE

We collected queries from the following three datasets to ensure coverage across multiple domains:

Table 7: Overview of Ranking-TAGER

| Dataset | Task Type | Cases | User Num | AVG. Length |
|---|---|---|---|---|
| Ranking-TAGER-RW | Routine Writing | 46,792 | 376 | 8637.37 |
| Ranking-TAGER-SG | Story Generation | 11,913 | 158 | 7525.57 |
| Ranking-TAGER-IF | Instruction Following | 20,312 | 335 | 7606.20 |

(1) **IMPACT**(Chia et al., 2023): This dataset contains 200 human-created prompts, 50 for each of the 4 diverse usage scenarios (Informative Writing, Professional Writing, Argumentative Writing, and Creative Writing), to evaluate LLMs' routine writing ability.

(2) **WritingPrompts**(Fan et al., 2018): This is a large dataset of 300K human-written stories paired with writing prompts from an online forum. We utilize the writing prompts part of the dataset to evaluate LLMs' story generation ability.

(3) **IFEval**(Zhou et al., 2023): This dataset contains 500+ prompts. The prompts include instructions such as "write an article with more than 800 words" and "wrap your response with double quotation marks". It evaluates the instruction following ability of LLMs.

### E.3 OVERVIEW OF RANKING-TAGER-RW, RANKING-TAGER-SG, RANKING-TAGER-IF

An overview of Ranking-TAGER-RW, Ranking-TAGER-SG, and Ranking-TAGER-IF can be found in Table 7.

### E.4 THE DETAILS OF THE COLLECTION PROCESS OF RANKING-TAGER

The Ranking-TAGER dataset was collected during our experimental process. We gathered interaction histories from 862 users with different preferences and processed them into the required dataset format. We cleaned and organized the collected data (e.g., removing interactions where the annotations provided by the AI Annotator did not meet the format requirements), ultimately resulting in the Ranking-TAGER dataset.

## F IMPLEMENTATION DETAILS

### F.1 HYPERPARAMETER CONFIGURATION

We set the following hyperparameters during the training process:
- $k$: The number of tags in the User Tag Set and the Tag Set Candidate.
- $m$: The total number of selected Tag Set Candidates for each query.
- $m_r$: The number of randomly selected Tag Set Candidates for each query.
- $s$: The size of Data Replay Buffer.
- $epoch$: The total number of epochs during the training process.
- $d$: The number of new Interaction Histories added to the Data Replay Buffer at the start of each epoch (for the first epoch, we add $s$ Interaction Histories to fill the Data Replay Buffer).
- $bnum$: The number of batches in each epoch.
- $bsz$: The number of data in each batch.
- $lr$: Learning rate.
- $\delta$: Weight of Auxiliary Entropy Loss.

Table 8: Hyperparameter Configurations for setups.

| | k | m | m_r | s | epoch | d | bnum | bsz | lr | delta |
|---|---|---|---|---|---|---|---|---|---|---|
| **3/20** | 3 | 3 | 2 | 25 | 15 | 3 | 8 | 20 | 2e-4 | 0 |
| **3/50** | 3 | 3 | 2 | 25 | 30 | 3 | 8 | 20 | 2e-4 | 4e-3 |
| **3/100** | 3 | 3 | 2 | 50 | 20 | 6 | 10 | 40 | 2e-4 | 4e-3 |

Our specific hyperparameter configurations for the "3/20," "3/50," and "3/100" setups are shown in Table 8. Additionally, we use torch.optim.AdamW as our optimizer during training, with all parameters set to their default values except for the learning rate.

### F.2 DATA SPLITS

For Ranking-TAGER-RW, our training set involves 120 different prompts from IMPACT (Chia et al., 2023), while both the test set and validation set contain 40 prompts each.

For Ranking-TAGER-SG, our training set involves 200 different prompts from WritingPrompts (Fan et al., 2018), while both the test set and validation set contain 40 prompts each.

For Ranking-TAGER-IF, our training set involves 200 different prompts from IFEval (Zhou et al., 2023), while both the test set and validation set contain 40 prompts each.

## G ADDITIONAL EXPERIMENTAL RESULTS

### G.1 DETAILED EXPERIMENTAL RESULTS ON INTERACTION COSTS

Table 9: The table presents the initial accuracy, accuracy after training, and the multiplier of improvement observed across multiple rounds of experiments. These experiments were conducted using only 30 interaction data points within the "3/50" setup of the Rabking-TAGER-RW dataset.

|  | Init. Accuracy (%) | Accuracy (%) | Multiplier |
|---|---|---|---|
| Run 1 | 2.39 | 18.11 | 7.58 |
| Run 2 | 9.18 | 15.69 | 1.71 |
| Run 3 | 5.42 | 29.67 | 5.47 |
| Run 4 | 4.48 | 44.26 | 9.88 |
| Run 5 | 7.88 | 32.45 | 4.12 |
| Run 6 | 16.13 | 54.31 | 3.37 |
| Run 7 | 4.15 | 13.09 | 3.15 |
| Run 8 | 3.67 | 15.32 | 4.17 |
| Run 9 | 3.96 | 10.52 | 2.66 |
| Run 10 | 6.34 | 14.22 | 2.24 |
| Average | 6.36 | 24.76 | 3.91 |

To validate that PREMIUM-Embed incurs a low "interaction cost," we trained our model using only 30 interaction data points in the "3/50" setup. Detailed experimental results can be found in Table 9.

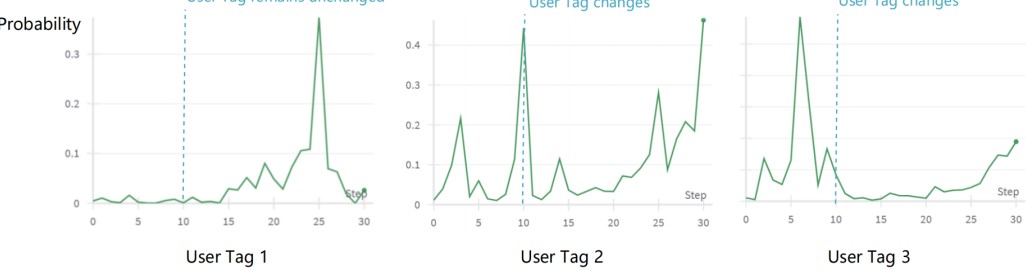

Figure 11: The probability of user tags being selected of dynamic user preferences.

### G.2 ADDITIONAL EXPERIMENTAL RESULTS FOR CASE STUDIES

**Dynamic User Preferences** Here, we provide additional experimental results for the experiment on "Dynamic User Preferences" in Section 6.2. Figure 11 shows the probability of each user tag being selected during training, corresponding to the experiment in Figure 3. In this experiment, user tag 1 remains unchanged, while user tags 2 and 3 are modified after 50 interactions between the user and the LLM.

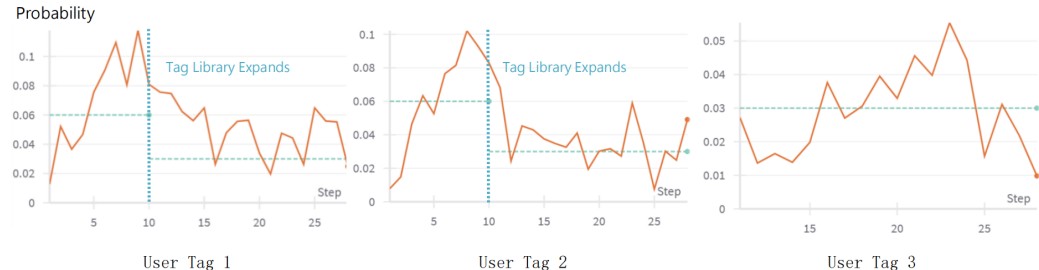

Figure 12: The probability of user tags being selected of expanded Tag Library. The horizontal dashed line represents 6 times the probability of tag selection under random selection, which decreases as the Tag Library expands due to the increase in tags in the Tag Library.

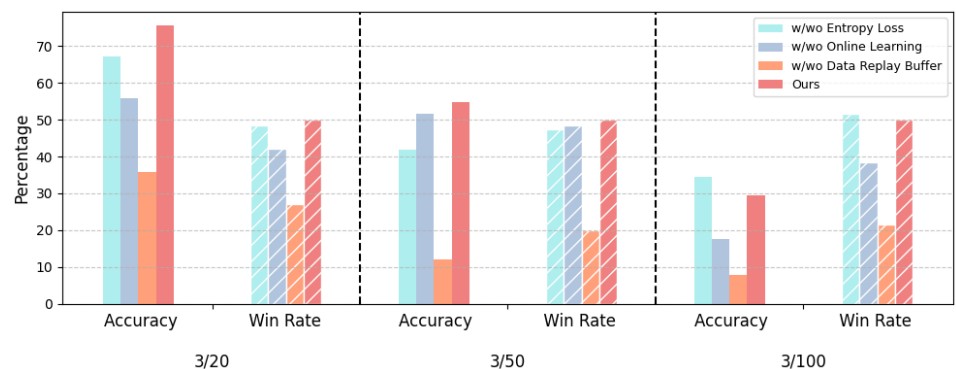

Figure 13: PREMIUM-Embed performs better than all the other variants on Ranking-TAGER-SG.

**Expanded Tag Library** Here, we provide additional experimental results for the experiment on "expanded Tag Library" in Section 6.2. Figure 12 shows the probability of each user tag being selected during training, corresponding to the experiment in Figure 4. In this experiment, user tag 1 and user tag 2 represent the user's initial preferences, while user tag 3 reflects the new user preference that emerges after the Tag Library expands at the 10th epoch of training.

### G.3 EVALUATION RESULTS OF ABLATION STUDY ON RANKING-TAGER-SG AND RANKING-TAGER-IF

Here, we report the evaluation results of ablation study on Ranking-TAGER-SG and Ranking-TAGER-IF, in Fig. 13 and Fig. 14, respectively. It is clear from these comparisons that our method outperforms all the variants in most setups and metrics, demonstrating the validity of our training approach.

## H FURTHER EXPERIMENTS

### H.1 PREMIUM IS SUITABLE FOR LLMS OF VARIOUS SIZES AND ARCHITECTURES

Our proposed PREMIUM framework is designed to be LLM-agnostic, working with both white-box and black-box LLMs. To demonstrate the versatility of the PREMIUM framework, we conduct additional comparative experiments using LLaMA-2 Chat (13B) and Qwen 1.5 Chat (32B) on the Ranking-TAGER dataset under the '3/50' setup. The detailed experimental results can be found in Table 10.

### H.2 HUMAN EVALUATION

In this work, we utilize AI annotation due to cost considerations, which has been widely adopted in recent research involving human feedback (Bai et al., 2022; Dubois et al., 2024; Lee et al., 2023). We anticipate that with human annotations providing more robust feedback consistency, the PREMIUM

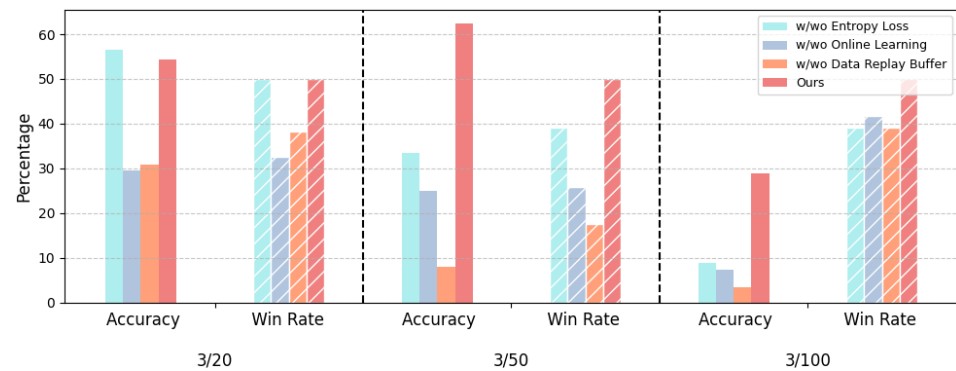

Figure 14: PREMIUM-Embed performs better than all the other variants on Ranking-TAGER-IF.

Table 10: **PREMIUM-Embed consistently outperforms all baselines across LLMs of various sizes and architectures as the backbone.** We conduct additional comparative experiments with LLaMA-2 Chat (13B) and Qwen 1.5 Chat (32B) on the Ranking-TAGER dataset using the '3/50' setup. Bold and underline denote the best and second-best results. All results are obtained by averaging the outcomes of multiple experiments. These experiments affirm the versatility of the PREMIUM framework.

| Dataset | Ranking-TAGER | | | | | |
|---|---|---|---|---|---|---|
| Subset | RW | | SG | | IF | |
| Metric | Accuracy | Win Rate | Accuracy | Win Rate | Accuracy | Win Rate |
| Backbone LLM | LLaMA-2 Chat(13B) | | | | | |
| Vanilla LLM | 6.00% | 20.00% | 6.00% | 11.25% | 6.00% | 33.75% |
| RALM | 8.45% | 27.50% | 6.10% | 13.75% | 5.52% | 45.00% |
| Population-Based Alignment | 19.23% | 33.75% | 34.93% | 40.00% | 6.69% | 35.00% |
| PREMIUM-Embed(Ours) | **44.42%** | **50.00%** | **49.58%** | **50.00%** | **41.95%** | **50.00%** |
| Backbone LLM | Qwen 1.5 Chat(32B) | | | | | |
| Vanilla LLM | 6.00% | 17.50% | 6.00% | 21.25% | 6.00% | 41.77% |
| RALM | 4.75% | 17.50% | 5.83% | 22.50% | 7.04% | 39.30% |
| Population-Based Alignment | 17.07% | 33.75% | 9.70% | 32.50% | 21.24% | 44.39% |
| PREMIUM-Embed(Ours) | **51.03%** | **50.00%** | **35.51%** | **50.00%** | **32.53%** | **50.00%** |

framework could achieve even better results, including fewer interaction requirements and more accurate alignment with user preferences.

To validate the effectiveness of our method in the face of real human preference feedback, we conduct small-scale human evaluation experiments. Specifically, we perform comparative experiments with five human users on the Ranking-TAGER-RW dataset using the "3/20" setup, which require only 45 interactions between users and the framework. The results, detailed in Table 11, demonstrate that our method achieves superior alignment with user preferences compared to all baselines, including OPPU and TidyBot. This underscores the effectiveness of the PREMIUM framework in practical applications.

### H.3 PREMIUM CAN EFFICIENTLY ALIGN WITH USER PREFERENCES EVEN WITH PAIRWISE FEEDBACK

In our experiments, we use three-choice ranking feedback to reduce the number of feedback instances required. This type of feedback is significantly easier to obtain compared to more complex forms, such as user edit feedback used by Gao et al. (2024) in PRELUDE and the ground truth personalized responses employed by Tan et al. (2024) in OPPU.

To demonstrate that our method can even accommodate simpler forms of preference feedback, we conduct experiments using pairwise comparison feedback instead of three-choice ranking feedback. This pairwise comparison feedback is easier to obtain and is commonly employed to capture human preference signals (e.g., DPO, IPO, SLiC). Our experimental results, detailed in Table 12, indicate that even with pairwise feedback, our framework can efficiently align with user preferences.

Table 11: **PREMIUM-Embed achieved more accurate preference alignment in human evaluation compared to other baselines.** Bold and underline denote the best and second-best results. Win rate compares each method's response with PREMIUM-Embed, with higher values indicating better performance. This demonstrates the effectiveness of the PREMIUM framework in practical applications and validates the feasibility of PREMIUM-Embed in real-world scenarios.

| Dataset | Ranking-TAGER-RW | | | | | |
|---|---|---|---|---|---|---|
| Metric | Win Rate | | | | | |
| Users | No.1 | No.2 | No.3 | No.4 | No.5 | Average |
| Vanilla LLM | 5.00% | 0.00% | 7.50% | 0.00% | 2.50% | 3.00% |
| RALM | 15.00% | 25.00% | 12.50% | 22.50% | 30.00% | 21.00% |
| TidyBot | 17.50% | 32.50% | 27.50% | 10.00% | 12.50% | 20.00% |
| OPPU(k=2) | 25.00% | 40.00% | 32.50% | 17.50% | 10.00% | 25.50% |
| PREMIUM-Embed (Ours) | 50.00% | 50.00% | 50.00% | 50.00% | 50.00% | **50.00%** |

Table 12: **PREMIUM-Embed efficiently aligned with user preferences across all datasets using only pairwise comparison feedback provided by users.** Bold denotes the best results. All results are obtained by averaging the outcomes of multiple experiments. All experiments were conducted using the "3/50" setup, with pairwise comparison feedback replacing three-choice ranking feedback.

| Dataset | Ranking-TAGER | | |
|---|---|---|---|
| Subset | RW | SG | IF |
| Method\Metric | Win Rate | | |
| Vanilla LLM | 6.00% | 6.00% | 6.00% |
| RALM | 9.68% | 5.23% | 9.43% |
| PREMIUM-Embed(Ours) | **44.28%** | **39.64%** | **23.20%** |

## H.4 PREMIUM-EMBED CAN EFFICIENTLY ADAPT TO DYNAMIC USER PREFERENCES UNDER MORE COMPLEX SETTINGS.

In the experiment on dynamic user preferences presented in Section 6.2, we demonstrate that PREMIUM-Embed can adapt to a dynamically changing User Tag Set. Here, we further showcase PREMIUM-Embed's ability to adapt to dynamic user preferences under a more complex setting.

Specifically, based on the original experimental setup, we introduce four groups of "binary tags" as discussed in Section 6.2. After 50 interactions between the user and the LLM, we modify two tags in the original User Tag Set and simultaneously change two binary tags in the Binary User Tag Set. Then, we allow the user with the updated preferences to continue interacting with the LLM. The experimental results, as shown in Figure 15, demonstrate that PREMIUM-Embed successfully adapts to the new user preferences with only 30 additional interaction data points, illustrating that PREMIUM-Embed can efficiently adapt to dynamic user preferences even under more complex settings.

## H.5 PREMIUM-EMBED DEMONSTRATES EFFECTIVE LLM PERSONALIZATION UNDER FEEDBACK PROVIDED BY DIFFERENT AI ANNOTATORS

To evaluate the impact of using different LLMs as AI Annotators on PREMIUM's performance, we present a comparative experiment involving various AI Annotators. Specifically, in addition to Qwen1.5-72B, we employ Mixtral-8x7B-Instruct-v0.1 (46.7B) and Mixtral-8x22B-Instruct-v0.1 (141B) as AI Annotators. The experiments are conducted on the Ranking-TAGER-RW Dataset under the "3/20," "3/50," and "3/100" settings, and the results are presented in Table 13.

The results indicate that regardless of the AI Annotator used, PREMIUM-Embed consistently demonstrates efficient alignment with user preferences. Furthermore, we observe that as the size of the AI Annotator model increases (which typically indicates stronger alignment with human capabilities), the personalization performance of PREMIUM-Embed improves. This suggests that with annotations providing more robust feedback consistency, the PREMIUM framework is capable of achieving better results.

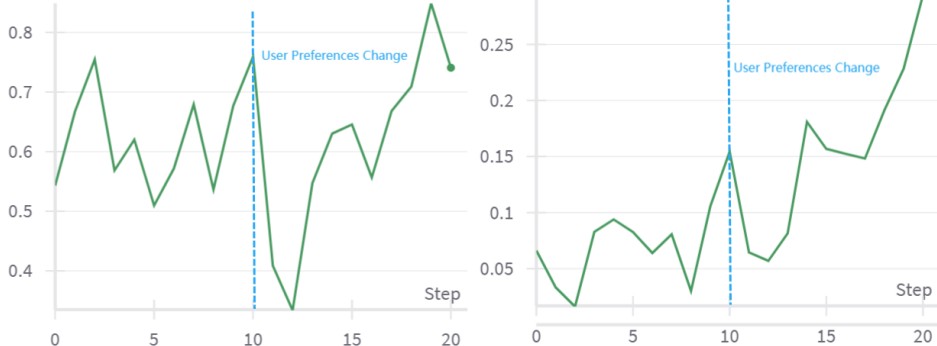

Figure 15: **PREMIUM-Embed effectively adapts to dynamic user preferences in a more complex setting involving both ordinary and binary tags.** The left figure shows the accuracy of binary tag selection, while the right figure presents the accuracy of ordinary tag selection. Within 30 interactions after the user preferences changed, PREMIUM-Embed improves both the ordinary and binary tag accuracies beyond their levels prior to the change in user preferences.

Table 13: **PREMIUM-Embed consistently demonstrates efficient alignment with user preferences regardless of the AI Annotator used.** The "Initial Accuracy" in the table represents the accuracy under random selection, serving as a reference. We use Mixtral-8x7B-Instruct-v0.1 (46.7B), Qwen1.5-72B, and Mixtral-8x22B-Instruct-v0.1 (141B) as AI Annotators. The results show that as the size of the AI Annotator model increases, the personalization performance of PREMIUM-Embed improves. Bold and underlined text denotes the best and second-best results, respectively.

| Dataset | Ranking-TAGER-RW | | |
|---|---|---|---|
| Setup | 3/20 (67 Cases) | 3/50 (112 Cases) | 3/100 (164 Cases) |
| AI Annotator\Metric | Accuracy | | |
| Initial Accuracy | 15.00% | 6.00% | 3.00% |
| Mixtral-8x7B (46.7B) | 49.65% | 32.05% | 30.32% |
| Qwen1.5 (72B) | 54.32% | 55.77% | 35.23% |
| Mixtral-8x22B (141B) | **64.29%** | **63.20%** | **47.44%** |

Table 14: **As the number of responses to be ranked increases, the personalization performance of PREMIUM-Embed improves.** The "Initial Accuracy" in the table represents the accuracy under random selection, serving as a reference.

| Dataset | Ranking-TAGER-RW | | | |
|---|---|---|---|---|
| Setup | 3/50 (112 Cases) | | | |
| Metric\Response Num | Init Accuracy | m=2 | m=3 | m=4 |
| Accuracy | 6.00% | 44.10% | 55.77% | 64.18% |

### H.6 EXPLORING THE IMPACT OF THE NUMBER OF RANKED RESPONSES ON THE PERFORMANCE OF PREMIUM-EMBED

Essentially, the parameter updates for PREMIUM-Embed rely on pairwise preference data extracted from preference ranking feedback. Therefore, the larger the number of response candidates $m$, the more data a single user feedback can provide for updating the tag selector.

In this section, we briefly explore the impact of the number of ranked responses on the performance of PREMIUM-Embed by testing with different values of $m$ (the number of responses to be ranked) as 2, 3, and 4 in the "3/50" setup of the Ranking-TAGER-RW Dataset. The results, shown in Table 14, indicate that as $m$ increases, the personalization performance of PREMIUM-Embed improves. Nevertheless, even when the feedback type is pairwise feedback or three-choice ranking feedback, which are relatively easy to obtain (corresponding to $m=2$ and $m=3$, respectively), PREMIUM-Embed still achieves efficient LLM personalization.

## I DETAILS AND DISCUSSION OF THE PERSONALIZED MOVIE TAGGING TASK IN LAMP-2

In Section 6, we conduct comparative experiments of PREMIUM-Embed against several methods on the "Personalized Movie Tagging" task from the LaMP-2 Dataset (Salemi et al., 2024b). In the settings of this task, the methods are provided with a predefined tag pool and a user's historical tagging data for several movies, and are required to predict which tags the user would assign to movies in the test set.

This task essentially involves using a user's historical preference data (choices related to product attributes) to predict their decisions and judgments on unseen items, which aligns with the essence of recommendation systems. Thus, validating the potential of the PREMIUM Framework in recommendation-related tasks.

## J THE REASONS AND ADVANTAGES OF CHOOSING OPEN-SOURCE LLMS AS THE BACKBONES FOR PREMIUM

In our experiments, we use Mistral-7B, LLaMA-2 Chat (13B), and Qwen-1.5 Chat (32B) as the backbones for PREMIUM. We see several key advantages in employing open-source LLMs:

1. **Proprietary LLMs often undergo frequent parameter updates, and their black-box nature poses challenges for result reproducibility.** Open-source LLMs eliminate these limitations, ensuring consistency in experimental setups.
2. **We strongly advocate for supporting the spirit of open source in both academia and industry.** Conducting experiments with open-source models not only aligns with this principle but also reflects the prevailing trend in academic research.

## K LIMITATION

In this work, the tags we used primarily describe user interests. However, a comprehensive user profile should also encompass other dimensions such as personality traits. Therefore, a promising future research direction is to utilize the Tagging System to capture a broader range of user attributes, aiming to achieve a more nuanced and in-depth alignment between LLMs and user preferences.

## L BROADER IMPACTS

Here, we discuss the broader impacts of this work. Our research aims to propose a novel LLM-agnostic framework for LLM personalization and introduces a lightweight, locally deployable implementation. The proposed PREMIUM framework enables both parameter-open LLMs (such as LLaMA-2) and black-box LLMs (such as GPT-3.5) to generate responses aligned with user preferences. This can be applied to a wide range of downstream tasks, encompassing customer service (Rome et al., 2024), personal health (Abbasian et al., 2024), and recommender systems (Li et al., 2024), demonstrating significant potential for positive societal impacts.

Moreover, our approach only requires users to provide Preference Ranking Feedback and does not necessitate any textual user information. The PREMIUM-Embed stores the learned user preferences within the neural network parameters rather than generating explicit textual user profiles, ensuring robust user privacy protection. To our knowledge, our work does not have any negative societal impacts.

## M ASSETS

### M.1 LICENSES FOR EXISTING ASSETS

**Datasets**
- **IMPACT** (Chia et al., 2023):

**License**: apache-2.0
**URL**: https://huggingface.co/datasets/declare-lab/InstructEvalImpact

•**WritingPrompts** (Fan et al., 2018):
**License**: MIT
**URL**: https://www.kaggle.com/datasets/ratthachat/writing-prompts

•**IFEval** (Zhou et al., 2023):
**License**: Unknown
**URL**:https://github.com/google-research/google-research/tree/master/instruction_following_eval

**Model**
**DRAGON-RoBERTa**:
**License**: CC-BY-NC 4.0
**URL**: https://github.com/facebookresearch/dpr-scale/tree/main/dragon

**LLMs**
• **Mistral-7B-Instruct-v0.2** (Jiang et al., 2023):
**License**: apache-2.0
**URL**: https://huggingface.co/mistralai/Mistral-7B-Instruct-v0.2

• **Qwen1.5-72B-Chat** (Bai et al., 2023):
**License**: tongyi-qianwen
**URL**: https://huggingface.co/Qwen/Qwen1.5-72B-Chat

## M.2    NEW ASSETS

We will later provide open access to the data and code, along with sufficient instructions to faithfully reproduce the main experimental results. This will include detailed documentation of the new assets we introduce (the code and the Ranking-TAGER dataset).

