# OpenReview forum: "PREMIUM: LLM Personalization with Individual-level Preference Feedback"
_ICLR.cc/2025/Conference — Submitted to ICLR 2025_

### Official Review · Reviewer_CwzU · 2024-10-30

**Soundness:** 3
**Presentation:** 4
**Contribution:** 3
**Rating:** 5
**Confidence:** 3

**Summary:**

This work developes a lightweight and dynamic framework PREMIUM, which capitalizes on the tag-based approaches in recommendation, to achieve alignment between LLMs and user preferences. Tags can be selected based on prompt-driven LLM generator or embedding matching. Also, it collects a AI-annotated dataset, Ranking-TAGER, specifically designed for evaluating LLM Personalization. Experiments are conducted on LaMP-2 and Ranking-TAGER.

**Strengths:**

1. It seems as a simple and lightweight framework, with significant improvement shown in the experiments.
2. It focuses on an interesting and important research question, Personalized LLM, with a well-surveyed understanding of the current state of the field.

**Weaknesses:**

1. One of the contributions, the construction of the Ranking-TAGER dataset, appears to be confusing. In my view, incorporating AI annotators is not persuasive. Also, Including tags as user characteristics in datasets is uncommon in real-world scenarios. Could you explain more about the contribution of Ranking-TAGER?
2. The evaluation setting of adapting to dynamic user preference, as in one of the challenges addressed, seems simple, because the changes in user preferences have already been abstracted by different tags. Could you provide experimental results on more difficult settings?

**Questions:**

1. Please see weakness.
2. A general question: What is the precise personalization that this work focuses on? Please provide a concise concept definition of "personalization" with specific examples.

---

> ### Author Response · Authors · 2024-11-21
>
> **Addressing Weakness 1: The Contributions of Ranking-TAGER**
>
> We appreciate the reviewer's feedback and for raising the confusion regarding Ranking-TAGER.
>
> First, as mentioned in our paper ***[L291-296]***, **the AI annotation used by Ranking-TAGER has been widely adopted in recent research involving human feedback** (such as Constitutional AI, Alpacafarm, and RLAIF), due to their high consistency with human annotators and significantly lower cost. Furthermore, the datasets labeled by AI Annotator do not contain any private information, thus **avoiding the privacy risks** that may arise from real-world datasets, as discussed in ***[L299-301]*** of the paper. Additionally, our dataset **includes the explanations provided by the AI annotator before offering the preference ranking. This makes our data highly interpretable and supports deeper analysis** ***[L304-307]***.
>
> Combining the tag system with AI annotation offers several benefits. For instance, **it allows the personalization capability of the methods to be more directly quantified through “accuracy,”** providing a valuable benchmark. Furthermore, since such data is difficult to obtain in real-world scenarios, we used AI annotations to **collect preference data from up to 862 different users simulated based on the tag system, totaling 79,017 preference data points**, as noted in ***[L300-302]*** of the paper. This further highlights the uniqueness and value of Ranking-TAGER.
>
> **Addressing Weakness 2: Additional Experiments which Increase the Complexity of the Experimental Settings for Dynamic User Preferences**
>
> We appreciate the reviewer's suggestion to increase the complexity of the experimental settings for dynamic user preferences.
>
> In PREMIUM's framework, the user profile is represented by the User Tag Set ***[L125-128]***, making it natural and reasonable to model changes in user preferences through modifications to the User Tag Set. Moreover, in the dynamic user preference experiment, **two out of three tags in the User Tag Set are changed**, as shown in  ***[L459-461]***, representing a significant shift in user preferences. We consider adapting to such new user preferences to be a challenging task.
>
> Following the reviewer's suggestion, we have added an experiment under a more difficult setting in ***[Appendix H.4]***. Based on the original experimental setup, **we introduce four groups of "binary tags" as discussed in our paper** ***[L475-482]***. After 50 interactions between the user and the LLM, **we modify two tags in the original User Tag Set and simultaneously change two binary tags in the binary User Tag Set.** Then, we allow the user with the updated preferences to continue interacting with the LLM. The experimental results, as shown in ***[Figure 15]***, demonstrate that **PREMIUM successfully adapts to the new user preferences with only 30 additional interaction data points**, illustrating that PREMIUM-Embed can efficiently adapt to dynamic user preferences even under more complex settings.
>
> **Addressing Question 2: The Definition of "LLM Personalization" in Our Work**
>
> We appreciate the reviewer’s comments regarding the confusion surrounding our definition of "LLM Personalization" in our work.
>
> In our submission, we introduced the objective of LLM personalization in ***[L144-150]*** under "Objective of Responses Generation": Our aim is to make the responses generated by LLMs **(1) relevant to the domains the user is interested in, and (2) adhere to the user’s instructions and answer the user’s questions.** For example, if a user interested in "Nutrition" asks, "How to make handmade desserts?", our goal is to select the tags "Nutrition" and "Bakery" from the tag library to assist the LLM in generating a response such as "To make handmade desserts with a focus on nutrition, consider using whole grain flour, natural sweeteners, and healthy fats…."
>
> Following the reviewer’s suggestion, we have added this example in ***[L147-150]*** to better illustrate our personalization goals. We believe this will help readers gain a clearer understanding of our objectives.

---

> ### Author Response · Authors · 2024-11-24
> **Could you let us know if our rebuttal has sufficiently addressed your concerns?**
>
> Dear Reviewer CwzU,
>
> We recognize that the timing of this discussion period may not align perfectly with your schedule, yet we would greatly value the opportunity to continue our dialogue before the deadline approaches.
>
> We hope that our responses and additional experiments have effectively addressed your concerns. We truly appreciate all the valuable advice we have received. **Could you let us know if your concerns have been adequately addressed? If you find that your concerns have been resolved, we would appreciate it if you could reconsider the review score.**
>
> Thanks!

---

> ### Author Response · Authors · 2024-11-27
> **Looking Forward to Further Discussion**
>
> Dear Reviewer CwzU,
>
> As the discussion period has now lasted for two weeks, we wanted to follow up and check if our responses have fully addressed your questions. Guided by your insightful comments, we have clarified several key points, including:
> - The contributions of Ranking-TAGER,
> - The definition of "LLM personalization" within the context of our work.
>
> Furthermore, we have conducted additional experiments to evaluate:
> - The ability of PREMIUM-Embed to efficiently adapt to dynamic user preferences in more complex settings.
>
> **Thank you again for your valuable feedback and suggestions, which have been instrumental in improving our paper. We look forward to your response.**
>
> Best regards,
> The Authors

---

> ### Author Response · Authors · 2024-11-30
> **Looking forward to your reply**
>
> Dear Reviewer CwzU,
>
> We would like to express our sincere appreciation for your positive opinions and constructive review of our paper on the occasion of Thanksgiving. We apologize for intruding during your busy schedule, but as the discussion period is near its end, **we would like to ensure our response aligns with your expectations and addresses your concerns. If you find that your concerns have been resolved, we would appreciate it if you could reconsider the review score.**
>
> Wishing you a joyful Thanksgiving,
>
> Best regards,
>
> Authors

---

### Official Review · Reviewer_j6HP · 2024-11-03

**Soundness:** 4
**Presentation:** 3
**Contribution:** 3
**Rating:** 6
**Confidence:** 4

**Summary:**

This paper proposed a new framework called PREMIUM for LLM personalization with individual-level preference. The framework adopts a tag system to represent user profiles and preferences, thereby guiding the LLM to generate more user-specific responses. The framework encompasses two main variants: PREMIUM-Prompt and PREMIUM-Embed, with the latter being particularly emphasized due to its efficiency and lightweight design, enabling deployment even on devices with limited computational resources.

The paper also introduced a new dataset, Ranking-TAGER, explicitly designed to evaluate LLM personalization methods that employ a tag-based user profile system. The dataset adopts an AI annotator to generate preference feedback with explanations.

The results demonstrate that PREMIUM-Embed consistently outperforms a variety of baselines across different datasets and setups. Notably, it achieves higher accuracy and win rates compared to methods that do not use tag system or do not leverage preference feedback.

Further experiments provide a deeper dive into the capabilities of PREMIUM-Embed, showcasing its adaptability to dynamic user preferences, generalizability to expanded tag libraries, and extendability to binary tags.

**Strengths:**

The paper addresses a timely and important topic: LLM personalization.

The proposed PREMIUM framework is LLM-agnostic, broadening its applicability and increasing its relevance to the research community. It is also efficient and lightweight, enabling local deployment and making personalized LLMs more accessible.

The Ranking-TAGER dataset is a valuable contribution to the research community, providing a much-needed benchmark for evaluating LLM personalization methods that utilize tag-based user profiles.

The paper conducted extensive experiments and comprehensive analysis to showcase the proposed framework's effectiveness.

**Weaknesses:**

The method of embedding based tag recommendation is well-studied in recommendation system community.

The assumption of preference feedback limits its applicability in real-word use cases.

It is unclear how groundtruth user tag is generated in evaluation.

Lack of discussion on how to determine the tags in tag library.

**Questions:**

1. In many cases, the queries asked by users may not need any personalization at all, such as reasoning tasks, or knowledge related QA tasks. How do these extra tags affect the performance of non-personalized tasks, which ideally should not be affected?

2. Would it be possible to also add more evaluation from other LLMs instead of just Qwen1.5-72B, to see how different LLMs as autorater affect performance?

3. What is the impact of m responses in preference ranking?

4. The paper could be even stronger in evaluation, if it can also do some evaluation with proprietary LLMs such as ChatGPT or Gemini.

5. The paper assumes the assumption of preference feedback during interaction, which might limit the applicability in real-world use cases.  It might be worth some discussion as its limitation.

---

> ### Author Response · Authors · 2024-11-21
>
> **Addressing Weakness 1: Novelty of PREMIUM beyond Embedding-based Tag Recommendation**
>
> We appreciate the reviewer pointing out that tag systems are well-studied in the recommendation system community.
>
> However, to the best of our knowledge, **our work is the first to introduce the perspective of tag systems into the field of LLM personalization**. Additionally, our key insights include **utilizing preference ranking feedback generated during interactions to achieve personalization without requiring an explicit user profile**, which addresses concerns related to user privacy.
>
> **Addressing Weakness 2 & Question 5: Why Using Preference Feedback is Practical in Real-World Scenarios**
>
> First, we appreciate the reviewer for raising concerns about the ease of obtaining preference feedback.
>
> In our submission, **as demonstrated in** ***[Appendix C.2],***  **the preference ranking feedback we adopt has several advantages compared to signals used in previous works.** It is readily accessible, unbiased, protects user privacy, is relevant to users' queries, and can enhance data collection efficiency.
>
> Moreover, **using preference data collected from rankings is a standard practice in LLM alignment research and has been widely applied in the field** (e.g., InstructGPT, RRHF, PRO). In our experiments in ***[Section 6 and Appendix H.3]***, we employ two common and easily obtainable types of feedback: three-choice ranking feedback and pairwise feedback. The results demonstrate that **PREMIUM-Embed can efficiently align with user preferences using both types of feedback.** These feedback types are significantly easier to collect compared to more complex forms, such as user edit feedback used by Gao et al. in PRELUDE and the "ground truth personalized responses" employed by Salemi et al. in ROPG.
>
> Additionally, as shown in our paper ***[L417-423] and Appendix G.1***, **PREMIUM-Embed can achieve efficient alignment with user preferences using only a minimal interaction cost (only 30 interaction data points).** We believe this effectively addresses concerns about the difficulty of collecting ranking feedback for PREMIUM.
>
> **Addressing Weakness 3: How the User Tag Set is initialized**
>
> We appreciate the reviewer's question regarding the method for initializing the User Tag Set.
>
> In each of our experiments, we randomly select a User Tag Set from the Tag Library to initialize the user's preferences. To explore a broad range of possible user preferences, our Ranking-TAGER covers 862 users with different preferences, as mentioned in ***[L300-302]*** of the paper. Following the reviewer’s suggestion, we have added further details on the initialization of the User Tag Set in ***[L356-358]*** of the experimental setup.
>
> **Addressing Weakness 4: The Design of the Tag Library**
>
> In our paper, as shown in ***[Appendix A]***, we have developed a Tag Library consisting of 200 tags covering 20 different areas to encompass a wide range of possible user interests. However, since this paper is research-oriented and not aimed at creating a market-ready product, the design of the Tag Library is not the focus of our discussion.
>
> Besides, **in Section 6.2, "PREMIUM-Embed can generalize to expanded Tag Library," and in** ***Appendix C.1 [L862-863]***, **we provide both experimental and theoretical evidence that the PREMIUM framework can be applied to larger Tag Libraries** (potentially containing tens of thousands of tags). This ensures that PREMIUM has a solid foundation for practical use in real-world applications.
>
> **Addressing Question 1: How to Prevent Extra Tags from Affecting the Personalized Generation**
>
> We appreciate the reviewer's concern regarding the potential impact of extra tags on the generated results.
>
> In our experiments, we did not observe any significant interference from these extra tags affecting the personalization capability of PREMIUM. However, following the reviewer’s spirit, we think that a simple solution to address this concern is to add a filter to the tag selector in PREMIUM. Specifically, we use the non-personalized Dragon retriever to compute the dot product between the embeddings of the selected tags and the user query embedding. We then set a threshold to filter out selected tags that are not closely related to the user query, thus preventing these extra tags from affecting the personalized generation.
>
> We plan to incorporate these suggested refinements and provide more detailed information in the final camera-ready version of this paper.

---

> > ### Author Response · Authors · 2024-11-21
> >
> > **Addressing Question 2: Evaluate the Impact of Using Different LLMs as AI Annotators on PREMIUM's Performance**
> >
> > We appreciate the reviewer's suggestion to evaluate the impact of using different LLMs as AI Annotators on PREMIUM's performance.
> >
> > In our submission, we select Qwen1.5-72B-Chat as our AI Annotator due to its strong alignment with human capabilities. Additionally, ***[Appendix H.2]*** demonstrates that PREMIUM achieves efficient LLM personalization relative to other baselines when addressing real human user preferences.
> >
> > Following the reviewer's suggestion, **we add a comparative experiment in** ***[Appendix H.5]*** **using other LLMs as AI Annotators.** Specifically, in addition to Qwen1.5-72B, we employ Mixtral-8x7B-Instruct-v0.1 (46.7B) and Mixtral-8x22B-Instruct-v0.1 (141B) as AI Annotators. The experiments are conducted on the Ranking-TAGER-RW Dataset under the "3/20," "3/50," and "3/100" setups, and results are presented in ***[Table 13]*** in our paper. We also present the data in the following table, where the values represent Accuracy.
> >
> > |**&nbsp;&nbsp;&nbsp;&nbsp;&nbsp;&nbsp;&nbsp;&nbsp;&nbsp;&nbsp;&nbsp;&nbsp;Setup**                      | **3/20 (67 Cases)** | **3/50 (112 Cases)** | **3/100 (164 Cases)** |
> > |:--------------------------------:|:-------------------:|:--------------------:|:---------------------:|
> > | **Initial Accuracy**            | 15.00%              | 6.00%                | 3.00%                 |
> > | **Mixtral-8x7B (46.7B)**        | 49.65%              | 32.05%               | 30.32%                |
> > | **Qwen1.5 (72B)**               | _54.32%_            | _55.77%_             | _35.23%_              |
> > | **Mixtral-8x22B (141B)**        | **64.29%**          | **63.20%**           | **47.44%**            |
> >
> > The results indicate that **regardless of the AI Annotator used, PREMIUM-Embed consistently demonstrates efficient alignment with user preferences. Furthermore, we observe that as the size of the AI Annotator model increases (typically indicating stronger alignment with human capabilities), the personalization performance of PREMIUM-Embed improves.** This observation aligns with our statement noted in ***[Appendix H.2]***: **with annotations providing more robust feedback consistency, the PREMIUM framework could achieve better results** ***[L1295, L1329].***
> >
> > **Addressing Question 3: The Impact of the Number of Response Candidates**
> >
> > We appreciate the reviewer's question regarding the impact of the number of response candidates.
> >
> > Essentially, the parameter updates for PREMIUM-Embed rely on pairwise preference data extracted from preference ranking feedback. Therefore, the larger the number of response candidates $m$, the more data a single user feedback can provide for updating the tag selector (as discussed in ***[L237-242]***).
> >
> > In our submission, **we employ two common and easily obtainable types of feedback in** ***[Section 6 and Appendix H.3]*** **: three-choice ranking feedback and pairwise feedback**. These types of feedback are widely used in many notable works on LLM alignment (e.g., RLHF, DPO, RRHF). Our experiments demonstrate that **PREMIUM-Embed can efficiently align with user preferences using both types of feedback.**
> >
> > Follow the reviewer's suggestion, **we conduct additional experiments on the number of responses to be ranked, which have been included in** ***[Appendix H.6].*** Specifically, we test different values of  $m$ (the number of responses to be ranked) as 2, 3, and 4 in the "3/50" setup of the Ranking-TAGER-RW Dataset. The results, presented in ***[Table 14]*** （Additionally, we present the data as shown below.）, demonstrate that **as m increases, the personalization performance of PREMIUM-Embed improves.** Nevertheless, even when the feedback type is pairwise feedback or three-choice ranking feedback—relatively easy to obtain (corresponding to  $m=2$ and  $m=3$, respectively)—PREMIUM-Embed still achieves effective LLM personalization.
> >
> > |  | **Init Accuracy**      | **&nbsp;&nbsp;m=2**                  | **&nbsp;&nbsp;m=3**                  | **&nbsp;&nbsp;m=4**                  |
> > |:------------------:|:------------------------:|:------------------------:|:------------------------:|:------------------------:|
> > | **Accuracy**      | 6.00%                    | 44.10%                   | 55.77%                   | 64.18%                   |

---

> > > ### Author Response · Authors · 2024-11-21
> > >
> > > **Addressing Question 4:  The Reasons for Choosing Open-source LLMs as the Backbones for PREMIUM**
> > >
> > > We appreciate the reviewer's suggestion regarding the evaluation with proprietary LLMs.
> > >
> > > In our submission, we use Mistral-7B, LLaMA-2 Chat (13B), and Qwen-1.5 Chat (32B) as the backbones for PREMIUM in ***[Section 6 and Appendix H.1]***, demonstrating that, **as an LLM-agnostic framework, PREMIUM is suitable for LLMs of various sizes and architectures**. We believe adapting PREMIUM to  proprietary LLMs such as ChatGPT or Gemini will be straightforward, if cost is not a concern.
> > >
> > > Moreover, there are several advantages to using open-source LLMs in our experiments:
> > >
> > > (1) **Many proprietary LLMs frequently update their APIs, and their black-box nature makes it challenging to reproduce results.** Using open-source LLMs avoids this issue.
> > >
> > > (2) We firmly believe that **both academia and industry should support the spirit of open source, and conducting experiments with open-source models aligns with the current trend in academic research.**
> > >
> > > Following the reviewer’s suggestion, we have added the reasons for choosing open-source LLMs as the backbones for PREMIUM to ***[Appendix J]***. We believe this will help readers better understand the benefits of our choice.

---

> > > > ### Comment · Reviewer_j6HP · 2024-12-02
> > > >
> > > > Thank you for taking the time to write the responses!
> > > >
> > > > I appreciate the authors give detailed responses to a few clarifying questions, which might be great to include in the paper or appendix for the following iterations. However, I'm still not fully convinced regarding:
> > > >
> > > > 1) the novelty of the embeddings based tag recommendation. A heavy chunk of the work is based on traditional embedding based approach to recommend tags, which is essentially a RAG-based approach with evaluation mostly focused on tag prediction accuracy instead of LLM capability in personalization. The performance of the approach relies heavily on the performance of tag selector (which might be the reason to propose exploration and exploitation tradeoff).
> > > >
> > > > 2) It still seems a pretty significant overhead/request to ask user to give preference feedback for about 30 rounds. For the user edit feedback used by Gao et al. in PRELUDE, the feedback is naturally generated by users making edits directly to the response, instead of nonorganic requests.

---

> > > > > ### Author Response · Authors · 2024-12-02
> > > > > **Response to Reviewer j6HP (1/2)**
> > > > >
> > > > > **Addressing Question 1: The Novelty of PREMIUM-Embed compared to Existing Methods**
> > > > > >
> > > > > > We appreciate the reviewer's focus on the novelty of embeddings-based tag recommendation.
> > > > > >
> > > > > > Existing RALM-based LLM personalization methods primarily focus on utilizing user textual information (e.g., interaction history) as retrieval sources (e.g., [1] [2] [3]). **However, such user textual information often introduces privacy risks and is challenging to obtain (for example, a new user in an application may not have any interaction history at all).** In contrast, the PREMIUM framework “retrieves” (selects) tags from the Tag Library, which does not rely on user-provided textual information. **This addresses both accessibility and privacy concerns, highlighting our novelty compared to existing RALM-based personalization methods.**
> > > > > >
> > > > > > In our submission, in addition to using the Accuracy of the tag selection as our metric, we also adopt the **Win Rate** metric to evaluate the personalization capability of our method by comparing its results against other baselines. This metric provides a more comprehensive assessment compared to Accuracy: **it not only considers the selection of user tags but also accounts for a more comprehensive range of factors influencing user preferences (such as the improved response quality resulting from selecting query-relevant tags)** ***[L343-348]***.  We report the comparison of our method with various baselines in terms of Win Rate under various experimental settings in ***[Table 2, Table 3, Table 10, Table 11, Table 12],*** **including results evaluated by AI Annotators as well as human evaluation results** ***[Table 11].*** The main results presented in ***[Table 2 and Table 3]*** demonstrate that **PREMIUM-Embed achieves a 2.5%-37.5% improvement in Win Rate compared to various baselines.** We believe this comprehensively and convincingly highlights the significant advantages of PREMIUM-Embed in enhancing LLM personalization capability.
> > > > > >
> > > > > > In addition, the Tag Selector is a key module of the PREMIUM framework. **PREMIUM initializes the Tag Selector using a powerful retriever with strong relevant information retrieval capabilities** ***[L312-313]*** **and fine-tunes it using user preference feedback** ***[L223-254]*** **to achieve the key goal of our LLM personalization: to make the responses (1) relevant to the domains the user is interested in, and (2) adhere to the user’s instructions and answer the user’s questions** ***[L144-151].*** Therefore, the careful design of the Tag Selector training (including proposing how to trade off between exploration and exploitation) and the use of the Accuracy metric to evaluate its personalization capability is both natural and reasonable.
> > > > > >
> > > > > > We plan to include the above discussion on the novelty of PREMIUM compared to other RALM-based methods, as well as insights into the design of the Tag Selector, in the appendix of the potential camera-ready version. We believe this will help readers gain a clearer understanding of the novelty of PREMIUM compared to existing methods. Additionally, we hope the reviewer can holistically consider our method's novelty and raise the score accordingly.
> > > > >
> > > > > **Addressing Question 2: PREMIUM has advantages in terms of data accessibility and privacy security and achieves higher data efficiency compared to alternative methods**
> > > > > > We appreciate the reviewer's focus on the issue of the difficulty in obtaining preference ranking data.
> > > > > >
> > > > > > We believe the discussion in our submission at ***[L916-917, L940-947]*** adequately addresses the reviewer's concerns. Regarding the user edit feedback employed by Gao et al. in PRELUDE, **it is often difficult to obtain or results in significant biases.** For example, the user may not know what the best personalized answer to their query is. Furthermore, **introducing textual edits from users raises privacy concerns.** In contrast, **the ranking feedback used in PREMIUM is both readily accessible and unbiased.** Even if the user is unsure about the optimal personalized response to their query, they can still provide an unbiased ranking of their preferences among the two or three responses presented to them. Additionally, **this process does not require users to provide any textual input, thereby eliminating privacy risks.**
> > > > > >
> > > > > > Besides, requiring users to edit LLM responses introduces a substantial overhead, **as it essentially asks users to create a new response and implicitly provide pairwise preferences between the original LLM-generated response and the new one. In contrast, PREMIUM avoids the need for users to generate new responses and achieves efficient personalization using only pairwise feedback, as demonstrated in** ***[Appendix H.3 and Table 12].***

---

> > > > > > ### Author Response · Authors · 2024-12-02
> > > > > > **Response to Reviewer j6HP (2/2)**
> > > > > >
> > > > > > > Moreover, Gao et al. **used 200 rounds of user edit feedback in their experiments**, as discussed in the "Task" section of their paper ([Section 4.1]). **This introduces a significantly higher interaction cost compared to the 30 rounds of ranking feedback we used in our experiments** ***[Section 6.1, L417-423].*** On the other hand, ranking feedback has already been used in commercial LLM services (e.g., OpenAI's ChatGPT occasionally asks users to choose which of two responses they prefer).
> > > > > > >
> > > > > > >
> > > > > > > On the other hand, we would like to point out that **PREMIUM-Embed achieves higher data efficiency compared to alternative methods.** First, if certain users cannot provide personal data of at least 30 samples, this is insufficient to fine-tune a personalized LLM using PEFT-based methods or to construct an effective retrieval source for RALM-based methods. As demonstrated in Table 1 of ROPG [3], these methods typically require dozens, or even hundreds, of personalized data samples from the same user to construct a retrieval source for various personalization tasks. Similarly, it is challenging for ICL-based methods to extract sufficiently complete user profiles from such limited data. For example, Christofer Richardson et al. [1] relies on summarizing dozens or even hundreds of user interaction records to construct a user profile for various personalization tasks, as shown in Table 1 of [4], which proposed the dataset used in [1].
> > > > > > >
> > > > > > > In contrast, PREMIUM offers higher data utilization efficiency. The experimental results shown in our paper ***[Tables 2 and 3]*** validate this point: **with the same amount of available user data, PREMIUM-Embed outperforms other baselines in various tasks and settings.** We believe this is because **PREMIUM models the personalization task as learning a small tag selector from preference feedback, which is inherently more data-efficient compared to fine-tuning LLMs on user data or using large amounts of user history as a retrieval source or for user profiling.**
> > > > > > >
> > > > > > > We plan to add the above discussion to the appendix of the potential camera-ready version, as we believe it will help readers better understand the advantages of preference ranking feedback compared to alternative solutions. At the same time, we hope the reviewer will holistically consider our contributions and raise the score.
> > > > > > >
> > > > > > > [1] Integrating Summarization and Retrieval for Enhanced Personalization via Large Language Models
> > > > > > >
> > > > > > > [2] Democratizing Large Language Models via Personalized Parameter-Efficient Fine-tuning
> > > > > > >
> > > > > > > [3] Optimization methods for personalizing large language models through retrieval augmentation
> > > > > > >
> > > > > > > [4]  Lamp: When large language models meet personalization

---

> ### Author Response · Authors · 2024-11-24
> **Could you let us know if our rebuttal has sufficiently addressed your concerns?**
>
> Dear Reviewer j6HP,
>
> We recognize that the timing of this discussion period may not align perfectly with your schedule, yet we would greatly value the opportunity to continue our dialogue before the deadline approaches.
>
> We hope that our responses and additional experiments have effectively addressed your concerns. We truly appreciate all the valuable advice we have received. **Could you let us know if your concerns have been adequately addressed? If you find that your concerns have been resolved, we would appreciate it if you could reconsider the review score.**
>
> Thanks!

---

> ### Author Response · Authors · 2024-11-27
> **Looking Forward to Further Discussion**
>
> Dear Reviewer j6HP,
>
> As the discussion period has now lasted for two weeks, we wanted to follow up and check whether our responses have fully addressed your questions. Guided by your insightful comments, we have clarified several key points, including:
> - The novelty of PREMIUM beyond embedding-based tag recommendation,
> - The practicality of using preference feedback,
> - A simple solution to prevent extra tags from affecting personalized generation,
> - The rationale for selecting open-source LLMs as backbones,
> - and more.
>
> Additionally, we have provided further experiments to evaluate:
> - The impact of using different LLMs as AI annotators,
> - The effect of the number of response candidates.
>
> **Thank you once again for your valuable comments and suggestions to help improve our paper. We look forward to hearing from you.**
>
> Best regards,
> The Authors

---

> ### Author Response · Authors · 2024-11-30
> **Looking forward to your reply**
>
> Dear Reviewer j6HP,
>
> We would like to express our sincere appreciation for your positive opinions and constructive review of our paper on the occasion of Thanksgiving. We apologize for intruding during your busy schedule, but as the discussion period is near its end, **we would like to ensure our response aligns with your expectations and addresses your concerns. If you find that your concerns have been resolved, we would appreciate it if you could reconsider the review score.**
>
> Wishing you a joyful Thanksgiving,
>
> Best regards,
>
> Authors

---

> ### Author Response · Authors · 2024-12-04
> **Kindly inquiry to revisit our responses and reconsider the score in light of the detailed clarifications provided**
>
> Dear Reviewer j6HP,
>
> We understand that the timing of this discussion period may not have perfectly aligned with your schedule, but we deeply appreciate the opportunity to engage with you. Thank you for your valuable feedback and suggestions.
>
> In our latest response, we have clarified the **novelty of PREMIUM-Embed compared to existing methods** and provided a detailed discussion on **how PREMIUM offers significant advantages in terms of data accessibility, privacy security, and data efficiency** compared to alternative approaches.
>
> We believe we have thoroughly addressed all of your concerns, and we hope these clarifications and additional insights have improved the quality of our work. **We would be truly grateful if you could review our response.** If you find that your concerns have been resolved, we kindly request that you reconsider the review score.
>
> Thank you once again for your thoughtful feedback and support.
>
> Best regards,
>
> The Authors

---

### Official Review · Reviewer_62bN · 2024-11-03

**Soundness:** 2
**Presentation:** 3
**Contribution:** 2
**Rating:** 5
**Confidence:** 4

**Summary:**

This paper proposes an LLM personalization method based on tag selection. Specifically, it introduces training a tag selector that adaptively selects suitable tags for each given query. The selected tags are then combined with the query to form a new prompt, which is used to request personalized responses from the LLM. The training of the tag selector relies on fitting user rankings of responses generated with different tags.

**Strengths:**

S1. The proposed method does not include training LLMs.
S2. The proposed method looks like it can be applied to closed-source LLMs.
S3. Most parts of the paper are generally easy to follow.

**Weaknesses:**

W1. Relying solely on tags for personalization may result in limited effectiveness, as the ability to personalize is heavily constrained by the predefined tags and their level of granularity. To my knowledge, the tag-based method is also not dominant for recommendation.

W2. The approach to achieving personalization is not clearly defined. User tags do not seem to be directly used for prompt generation but are instead used to rank the generated outputs. If this is the case, achieving personalization would require the tag selector to be specific to each user. This implies that a separate selector would need to be learned for each user, which could lead to efficiency issues and reduced learning efficacy, especially when data is sparse.

W3. Many important baselines, e.g., [1] and [2], are not compared.

W4. When applied to real-world scenarios, the method requires users to rank numerous responses, which could hinder its practical implementation.

W5. The recommendation appears to be a suitable task for evaluating personalization capabilities. Why not directly assess the method's effectiveness with the recommendation task?

W6. How does the method perform compared to soft-prompt tuning-based methods?

[1] Optimization Methods for Personalizing Large Language Models through Retrieval Augmentation.
[2] User Embedding Model for Personalized Language Prompting.

**Questions:**

See weaknesses.

---

> ### Author Response · Authors · 2024-11-21
>
> **Addressing Weakness 1: Concerns regarding Tag-Based Personalization**
>
> First, we appreciate the reviewer for raising concerns about using tags to characterize user preferences.
>
> We have addressed the advantages of using the tag system in PREMIUM in ***[Appendix C.1]*** **of our paper,** which we believe effectively addresses the reviewer’s concerns.
>
> **1. The granularity of the tag system can cover sufficient diversity among humans**:
> The tag system in PREMIUM offers sufficient diversity to capture user differences. For the "3/100" setup, the potential combinations represent up to 160k different user types. Furthermore, in ***[Section 6.2]***, we discuss the extension to binary tags, which can theoretically represent up to $2^{100}$ different user types. This ensures broad coverage of user diversity while maintaining linear scalability in compute and storage costs.
>
> **2. Superiority of the Tag System to Alternative Solutions**:
> Unlike traditional LLM personalization methods, such as PEFT or approaches like RAG and ICL that integrate textual user information, our tag system avoids issues with flexibility, privacy, and cost. Although tags may be less expressive, they provide significant advantages in terms of privacy protection, cost-efficiency, and scalability.
>
> **Addressing Weakness 2: How the Tag System Operates in PREMIUM and Concerns Regarding PREMIUM's Efficiency**
>
> We appreciate the reviewer for raising the confusion regarding the operation of the tag system in PREMIUM.
>
> In our submission, we describe how PREMIUM achieves LLM personalization in **Section 2, "Responses Generation through the Tagging System"** ***[L101-141], and Figure 1.*** PREMIUM utilizes a tag selector to choose tags that guide the LLM in generating responses aligned with the selected tags. **The user (AI Annotator) provides ranking feedback for the responses based on the User Tag Set, but PREMIUM itself does not have visibility into the User Tag Set.** This feedback is then used to fine-tune the tag selector to align with user preferences.
>
> Regarding the reviewer's concerns about PREMIUM's efficiency and learning efficacy, we believe that the experimental results in ***[L424-427] and Table 4*** address these concerns well: **Laptop-level resources are sufficient for PREMIUM-Embed.** **Moreover, current LLM personalization methods often require users to deploy and fine-tune local LLMs** (e.g., Baize, and the UEM mentioned by the reviewer), which generally incur very high computational costs (***[L42-46]***). In contrast, **PREMIUM only requires the user to deploy and fine-tune a tag selector locally, which is less than 1GB in size**, significantly reducing the computational cost burden for users.
>
> In addition, regarding the reviewer's concern about sparse data, the experimental results in ***[L417-423] and Appendix G.1*** demonstrate that **PREMIUM-Embed can achieve efficient alignment with user preferences using only minimal interaction cost (only 30 interaction data points).** We follow the established alignment research setting based on ranking feedback, which is widely used in many notable works (e.g., InstructGPT, DPO, RRHF). While such user feedback may be sparse, it remains an open problem for the research community and is not the focus of this paper.
>
> **Addressing Weakness 3 & Weakness 6: ROPG Proposed in [1], UEM Proposed in [2], and Other Soft-Prompt Tuning-Based Methods**
>
> We appreciate the reviewer's suggestion to compare PREMIUM with other methods, including soft-prompt tuning-based approaches. However, these methods are difficult to apply in real-world settings.
>
> Regarding ROPG proposed in [1], **it not only requires leveraging a textual User Profile but also asks users to provide  "ground truth personalized outputs" for their user queries to calculate the deviation from LLM responses** (details can be found in Section 3 of [1]). This approach **introduces privacy concerns and is inherently impractical, as users often cannot provide or may provide biased "ground truth personalized outputs."** We have discussed the inherent impracticality of this setting in our submission ***[L916-917, L940-943]***. Furthermore, the discussion in Section 5 and Figure 3 of [1] indicates that **ROPG did not demonstrate significant advantages over the baselines.**
>
> For UEM proposed in [2] and other soft-prompt tuning-based methods, they **require users to locally deploy and fine-tune the LLM. Although these methods are parameter-efficient, they are not computationally efficient.** Locally fine-tuning an LLM still imposes an unbearable burden on individual users, as we mentioned in ***[L42-46]***. For example, **individual users typically do not have access to computational resources equivalent to the v3-8 TPUs used in the experiments of [2].** Additionally, **the code for the method in [2] is not open-sourced**, which presents a major obstacle for reproducing their approach.

---

> > ### Author Response · Authors · 2024-11-21
> >
> > **Addressing Weakness 4: Why Using Preference Feedback is Practical in Real-World Scenarios**
> >
> > First, we appreciate the reviewer for raising concerns about the ease of obtaining preference feedback.
> >
> > In our submission, **as demonstrated in** ***[Appendix C.2],***  **the preference ranking feedback we adopt has several advantages compared to signals used in previous works.** It is readily accessible, unbiased, protects user privacy, is relevant to users' queries, and can enhance data collection efficiency.
> >
> > Moreover, **using preference data collected from rankings is a standard practice in LLM alignment research and has been widely applied in the field** (e.g., InstructGPT, RRHF, PRO). In our experiments in ***[Section 6 and Appendix H.3]***, we employ two common and easily obtainable types of feedback: three-choice ranking feedback and pairwise feedback. The results demonstrate that **PREMIUM-Embed can efficiently align with user preferences using both types of feedback.** These feedback types are significantly easier to collect compared to more complex forms, such as user edit feedback used by Gao et al. in PRELUDE and the "ground truth personalized responses" employed by Salemi et al. in ROPG.
> >
> > Additionally, as shown in our paper ***[L417-423] and Appendix G.1***, **PREMIUM-Embed can achieve efficient alignment with user preferences using only a minimal interaction cost (only 30 interaction data points).** We believe this effectively addresses concerns about the difficulty of collecting ranking feedback for PREMIUM.
> >
> > **Addressing Weakness 5:  Assessing our method's effectiveness with the recommendation task**
> >
> > We appreciate the reviewer's suggestion to assess our method's effectiveness with the recommendation task.
> >
> > In our submission, we conduct comparative experiments of PREMIUM-Embed against several methods on the **Personalized Movie Tagging task** from the LaMP-2 Dataset ***([L334-339] and Table 3)***. In the settings of this task, the methods are provided with a predefined tag pool and a user's historical tagging data for several movies, and are required to predict which tags the user would assign to movies in the test set. **This task essentially involves using a user's historical preference data (choices related to product attributes) to predict their decisions and judgments on unseen items, which aligns with the essence of recommendation systems.** Therefore, we consider it fundamentally a recommendation task.
> >
> > Following the reviewer's suggestion, we have added an introduction and discussion on the Personalized Movie Tagging task in ***[Appendix I]***, which we believe helps readers understand the rationale for selecting this task for evaluation and further highlights the personalization capabilities of the PREMIUM framework.

---

> ### Author Response · Authors · 2024-11-24
> **Could you let us know if our rebuttal has sufficiently addressed your concerns?**
>
> Dear Reviewer 62bN,
>
> We recognize that the timing of this discussion period may not align perfectly with your schedule, yet we would greatly value the opportunity to continue our dialogue before the deadline approaches.
>
> We hope that our responses have effectively addressed your concerns. We truly appreciate all the valuable advice we have received. **Could you let us know if your concerns have been adequately addressed? If you find that your concerns have been resolved, we would appreciate it if you could reconsider the review score.**
>
> Thanks!

---

> > ### Comment · Reviewer_62bN · 2024-11-25
> >
> > Thank you for your rebuttal. However, it seems that my concerns have not been fully addressed.
> >
> > W1: I still believe that relying solely on tags makes it difficult to capture user information.
> >
> > W2: My concern is that a selector needs to be trained for each user, but the data available for each user is often limited. This makes it challenging to train an effective selector. For instance, in real-world recommendation datasets, many users typically have fewer than 10 samples.
> >
> > W3 & W6: Asking users to provide feedback on tags actually carries a risk of privacy leakage. As mentioned in the response about Weaknesses 5, the task itself involves predicting which tags the user would assign to movies.
> >
> > W4: In the real world, asking users to provide 30 rounds of feedback imposes a substantial burden and may result in user attrition.
> >
> > W5: This is a tag prediction task. Did other baselines utilize user feedback on tags?

---

> > > ### Author Response · Authors · 2024-11-30
> > > **Response to Reviewer 62bN (1/2)**
> > >
> > > **Addressing Weakness 1: The advantages of the Tag System compared to alternative solutions**
> > > >Relying on User Tags to represent user profiles is somewhat less expressive than alternative solutions, such as integrating user information into model parameters or utilizing textual user data, as discussed in our submission ***[L908-910]***. However, as we mentioned in ***[L904-910]***, **the trade-off in expressiveness of the Tag System brings significant advantages in terms of cost efficiency, privacy security, and flexibility.** We hope the reviewer will holistically consider the advantages of the Tag System.
> > >
> > > **Addressing Weakness 2 & 4: PREMIUM achieves higher data efficiency compared to alternative methods**
> > >
> > > > We appreciate the reviewer's focus on the issue of the scarcity of personalized data for each user.
> > > >
> > > > We believe that, **compared to existing LLM personalization methods, PREMIUM achieves higher data efficiency.** For certain users who have fewer than 10 data samples, this is insufficient to fine-tune a personalized LLM using PEFT-based methods, or to construct an effective retrieval source for RALM-based methods (As demonstrated in Table 1 of ROPG [1], the method typically requires dozens, or even hundreds, of personalized data samples from the same user to construct retrieval source for various personalization tasks). Similarly, it is challenging for ICL-based methods to extract sufficiently complete user profiles from such limited data (For example, Christofer Richardson et al. [2] relies on summarizing dozens or even hundreds of user interaction records to construct a user profile for various personalization tasks，as shown in Table 1 of [3], which proposed the dataset used in [2]).
> > > >
> > > > In contrast, PREMIUM offers higher data utilization efficiency. The experimental results shown in our paper ***[Tables 2 and 3]*** validate this point: **with the same amount of available user data, PREMIUM-Embed outperforms other baselines in various tasks and settings.** We believe this is because **PREMIUM models the personalization task as learning a small tag selector from preference feedback, which is inherently more data-efficient compared to fine-tuning LLMs on user data or using large amounts of user history as a retrieval source or for user profiling.**
> > > >
> > > > [1] Optimization methods for personalizing large language models through retrieval augmentation
> > > >
> > > > [2] Integrating Summarization and Retrieval for Enhanced Personalization via Large Language Models
> > > >
> > > > [3] Lamp: When large language models meet personalization
> > >
> > > **Addressing Weakness 3 & 6: Ranking Feedback on LLM Responses offers advantages in privacy compared to alternative types of feedback**
> > >
> > > >To achieve LLM personalization tasks, any method requires obtaining signals of users' personal preferences. **For our proposed PREMIUM framework, it leverages users' preference rankings of LLM responses, rather than direct feedback on the selected tags themselves [L151-158].** Even in the Personalized Movie Tagging task we selected for our experiments, the user feedback we use is based on the movie tags within the LLM responses, not the tags chosen by the tag selector (which serve as the Reference Opinion), as demonstrated in the prompt presented below.
> > > >
> > > >In contrast to the textual user information or "ground truth personalized responses" used in existing methods, our approach relies on ranking feedback on responses, **which does not involve any user-provided text nor direct preferences for the tags in the Tag Library** (as discussed in ***[L943-947]***). This approach clearly provides advantages in terms of privacy and security.
> > > >
> > > >In line with the reviewer's spirit, we will include the prompt used in the Personalized Movie Tagging task in the potential camera-ready version. We believe this will help clarify the setup of our method for this task for the readers.
> > >
> > > ```
> > > Based on the movie description provided by the user and the given reference opinion, please determine which tag the movie relates to among the following tags. If the user's reference opinion is reasonable, your response should simply match the reference opinion; otherwise, choose the tag you believe is correct among the following tags. Just answer with the tag name without further explanation.
> > >
> > > tags: [sci-fi, based on a book, comedy, action, twist ending, dystopia, dark comedy, classic, psychology, fantasy, romance, thought-provoking, social commentary, violence, true story]
> > >
> > > The user's input is in this format:
> > > [Movie Description]
> > > { description }
> > > [Reference Opinion]
> > > { tag }
> > > Your answer must follow this format:
> > > {one of the given tags}
> > > ```

---

> > > > ### Author Response · Authors · 2024-11-30
> > > > **Response to Reviewer 62bN (2/2)**
> > > >
> > > > **Addressing Weakness 5: The implementation details of the Personalized Movie Tagging experiments and additional experiments on the recommendation task**
> > > >
> > > > >In the Personalized Movie Tagging task used for our experiments, **both OPPU and TidyBot utilize the {Movie Description - User Tag} pairs provided in LaMP-2** as retrieval sources or to summarize the user's interaction history for user profiling. In contrast, our PREMIUM framework relies solely on **an equal amount of ranking feedback for responses** (based on the ground truth user tags available in LaMP-2 ***[L337-339]***). For instance, if the ground truth tag is "sci-fi" and the three LLM-generated responses are "sci-fi," "comedy," and "action," the ranking feedback would be "sci-fi" > "comedy" > "action" or "sci-fi" > "action" > "comedy."
> > > > >
> > > > >**Compared to other baselines that can directly access ground truth user tags, the ranking feedback we use contains less personal user information (e.g., when none of the three responses contain the ground truth user tag).** This highlights both the efficient personalization capability of our method and its advantages in protecting user privacy.
> > > > >
> > > > >Following the reviewer's spirit, we will include the details of the Personalized Movie Tagging task in the potential camera-ready version. We believe this will help clarify the settings of this task for the readers and highlight the efficient LLM personalization capabilities of PREMIUM-Embed.
> > > >
> > > > >Moreover, following the reviewer's suggestion, **we assess the personalization capabilities of PREMIUM in the recommendation task using a subset of the Amazon Review Data (2018) dataset [4].** The task involves providing the LLM with the titles, descriptions, and categories of three items and asking it to recommend one to the user. Specifically, we use the "Movies and TV" data from the Amazon Review Data (2018) and select five active users based on the available number of reviews. For each user, we extract 135 reviews, each containing {item title, item description, item categories, and user rating}. These 135 reviews are split into a training set and a test set at a 2:1 ratio. The dataset includes 15 categories across all items, which we treat as the Tag Library for PREMIUM. We choose ICL-Based TidyBot [5] and OPPU [6], which use RALM and ICL-based personalization methods, as our baselines. **These methods leverage the review information of 90 items from the training set to generate user profiles or retrieval-augmented sources. In contrast, PREMIUM-Embed uses an equal number of ranking feedback responses, with feedback derived from item ratings (which are not visible to PREMIUM).** All items used during the PREMIUM training process are within the training set. **We use ‘Accuracy’ as the evaluation metric, i.e., the probability of successfully recommending the highest-rated item to the user.** The experimental results are presented in the table below: **compared to the baselines, PREMIUM-Embedding achieves a 16%-28% improvement in Accuracy, demonstrating its superior personalization performance in the recommendation task.**
> > > >
> > > >
> > > > **Comparative performance results on the Amazon Review Data (2018) dataset.** Bold text indicates the best results. $k$ represents the $top-k$ user histories provided to the LLM in the retrieval-augmented generation process. The $Random Select$ row shows the Accuracy achieved by randomly selecting items in the recommendation task, serving as a baseline reference.
> > > >
> > > > |      | **User 1** | **User 2** | **User 3** | **User 4** | **User 5** | **Average** |
> > > > |:-----:|:-------:|:-----------:|:----------------:|:-------------------:|:-------------------:|:---------------:|
> > > > | **Random Select**   | 33.33%  | 33.33%  | 33.33%  | 33.33%  | 33.33%  | 33.33% |
> > > > | **TidyBot** |  33.33%  | 33.33% | 33.33%  | 40.00% | 46.67% | 37.33% |
> > > > | **OPPU (k=1)**  | 46.67%   | 53.33%   | 26.67%  | 40.00% | 33.33% | 40.00% |
> > > > | **OPPU (k=2)**  | 26.67%   | 6.67% | 33.33%  | 26.67%| 46.67%| 28.00%|
> > > > | **OPPU (k=4)**  | 33.33%   | 20.00% | 33.33%   | 33.33% | 26.67%| 29.33%|
> > > > | **PREMIUM-Embed**  | **66.67%** | **53.33%** | **40.00%** | **60.00%** | **60.00%** | **56.00%** |
> > > >
> > > > >We will include a detailed description of this experiment, including dataset details, prompt templates used, training parameters, etc., in the potential camera-ready version. We believe this will help provide a more comprehensive evaluation and demonstrate the efficiency of PREMIUM in various personalization tasks.
> > > > >
> > > > >[4] Modeling the visual evolution of fashion trends with one-class collaborative filtering
> > > > >
> > > > >[5] Tidybot: personalized robot assistance with large language models
> > > > >
> > > > >[6] Democratizing Large Language Models via Personalized Parameter-Efficient Fine-tuning

---

> ### Author Response · Authors · 2024-12-02
> **A friendly reminder for further discussion**
>
> Dear Reviewer 62bN,
>
> We hope this message finds you well. The rebuttal phase ends today and we would like to know if our further response has completely addressed your concerns. We would really appreciate that if you could check our response. If you find that your concerns have been resolved, we would appreciate it if you could reconsider the review score. Looking forward to hearing back from you.
>
> Best Regards,
>
> Authors

---

> ### Author Response · Authors · 2024-12-04
> **Kindly inquiry to revisit our responses and reconsider the score in light of the detailed clarifications and experiments provided**
>
> Dear Reviewer 62bN,
>
> We understand that the timing of this discussion period may not have perfectly aligned with your schedule, but we deeply appreciate the opportunity to engage with you. Thank you sincerely for your valuable feedback and suggestions.
>
> In our further responses, we have provided:
> - A detailed discussion on **the advantages of the Tag System compared to alternative solutions**;
> - An explanation of how **PREMIUM achieves higher data efficiency compared to alternative methods**;
> - A clarification of the **advantages of Ranking Feedback on LLM Responses in terms of privacy** compared to alternative types of feedback;
> - The **implementation details of the Personalized Movie Tagging experiments**.
>
> In addition, we have conducted **extensive comparative experiments on the recommendation task using a subset of the Amazon Review Data (2018) dataset**. We dedicated significant time and effort to completing these experiments, which we believe address the concerns raised and further strengthen the quality of our paper.
>
> We hope these clarifications and additional contributions have resolved all your previous concerns. **We would greatly appreciate it if you could review our response and, if you find your concerns addressed, kindly reconsider the review score.**
>
> Thank you again for your thoughtful feedback and time.
>
> **Best regards,**
> The Authors

---

### Official Review · Reviewer_b5ce · 2024-11-04

**Soundness:** 3
**Presentation:** 3
**Contribution:** 3
**Rating:** 6
**Confidence:** 2

**Summary:**

The authors propose PREMIUM, a framework for LLM-agnostic personalization.\
PREMIUM uses a tag-based system inspired by personality typology and recommendation systems to capture user preferences.\
This approach, along with a variant called PREMIUM-Embed, can run efficiently on minimal hardware, such as a laptop.\
Extensive experiments show that PREMIUM significantly improves personalization accuracy and adaptability to changing user preferences.

**Strengths:**

1. The proposed method provides a well-structured approach to selecting user tags, which enhances the personalization process by aligning it with user-specific characteristics.
2. The introduction of a new benchmark dataset, Ranking-TAGER, specifically designed for personalized LLMs, is a valuable contribution. \
This dataset not only allows for standardized evaluation of personalized language models but also fills a critical gap in LLM research.

**Weaknesses:**

1. The explanation of the process after tag selection in Section 4 is unclear.\
While the paper describes selecting tags based on the embeddings of the query and tags, it lacks a clear follow-up on how these tags are subsequently used to influence the LLM's responses.\
A detailed, step-by-step breakdown of how the selected tags are integrated into the LLM's response generation would help clarify this process.\
Additionally, a flowchart or diagram showing the progression from tag selection to final response would improve comprehensibility.
2. A user study focusing on the readability and user perception of the tag selection process would enhance the paper's empirical rigor. \
Specifically, assessing the clarity of the tag-based personalization process and how users perceive its effectiveness in delivering personalized responses would provide valuable insights.

**Questions:**

Please refer to Weaknesses.

---

> ### Author Response · Authors · 2024-11-21
>
> **Addressing Weakness 1: How the Selected Tags in PREMIUM Assist the LLM in Generating Responses:**
>
> We appreciate the reviewer's feedback and for raising the confusion regarding how the selected tags in PREMIUM assist the LLM in generating responses.
>
> In our submission, we have described this process in ***Section 2: [Lines 135-143]***. Specifically, the PREMIUM framework **combines the user query with the selected tags into a prompt**, which is then fed into the LLM. **This prompt guides the LLM to generate the response that includes elements, perspectives, examples, and terminologies related to the selected tags.** The prompt template used in this process is shown in ***[Figure 6 of Appendix B]*** under the "Ordinary Setup." The combination of selected tags and user queries into prompts, as well as the process of feeding them into the LLM to assist response generation, is also illustrated on the right side of ***[Figure 1 in Section 2]***.
>
> Follow the reviewer’s spirit, to help readers more easily locate the prompt and understand the process, we have added a link to Appendix B in ***[Lines 139-143]*** of our submission. We believe this improvement will enhance the readability of our paper.
>
> **Addressing Weakness 2: How users perceive the effectiveness of PREMIUM in delivering personalized responses:**
>
> We appreciate the reviewer's focus on the issue of how users perceive the effectiveness of PREMIUM in delivering personalized responses.
>
> In our submission, we present a human evaluation involving multiple human users in ***[Appendix H.2 and Figure 11]***. The results indicate that when using human annotations that provide more robust feedback consistency, PREMIUM demonstrates strong personalization capabilities compared to multiple baselines. Specifically, **PREMIUM achieves a win rate that is 25%-47% higher than other baselines, with only 45 required interactions**. We believe this validates the real users' acknowledgment of the effectiveness of personalized responses provided by PREMIUM-Embed.

---

> ### Author Response · Authors · 2024-11-24
> **Could you let us know if our rebuttal has sufficiently addressed your concerns?**
>
> Dear Reviewer b5ce,
>
> We recognize that the timing of this discussion period may not align perfectly with your schedule, yet we would greatly value the opportunity to continue our dialogue before the deadline approaches.
>
> We hope that our responses have effectively addressed your concerns. We truly appreciate all the valuable advice we have received. **Could you let us know if your concerns have been adequately addressed? If you find that your concerns have been resolved, we would appreciate it if you could reconsider the review score.**
>
> Thanks!

---

> > ### Comment · Reviewer_b5ce · 2024-11-26
> >
> > Thank you for your response.\
> > After I had read your responses, I decided to raise my score to 6.

---

> > > ### Author Response · Authors · 2024-11-26
> > >
> > > Dear reviewer b5ce,
> > >
> > > Thank you for your thoughtful and constructive feedback. We are pleased to hear that our responses have addressed your concerns. We are committed to incorporating the suggested changes in our revisions to further enhance the manuscript.

---

### Author Response · Authors · 2024-12-04
**Summary of Additional Experiments and Clarifications from the Discussion Phase**

We sincerely thank the reviewers for their constructive feedback during the discussion phase and their recognition of the novelty, clarity, and comprehensive results of our work. We appreciate their acknowledgment of our paper’s contributions, including:
1. **Introducing a lightweight and efficient LLM-agnostic framework for LLM personalization**
2. **Establishing a much-needed benchmark for standardized evaluation of personalized language models**
3. **Demonstrating substantial performance improvements over diverse baselines across multiple datasets via extensive experiments with the PREMIUM framework**

In response to the concerns raised during the review and rebuttal process, we have conducted **extensive additional experiments** to further strengthen our submission:
-  An experiment which increase the complexity of the experimental settings for dynamic user preferences  ***[Appendix H.4 and Figure 15]***.
- Comparative experiments that evaluate the impact of using different LLMs as AI Annotators  ***[Appendix H.5 and Table 13]***
- An experiment to evaluate the impact of the number of response candidates ***[Appendix H.6 and Table 14]***
- Comparative experiments on the recommendation task using a subset of the Amazon Review Data (2018) dataset, as detailed in ***[our response to Reviewer 62bN]*** (to be added to the potential camera-ready version).

We have also **clarified several aspects** of our submission in the rebuttal and made significant improvements to the revised version of the paper. Key clarifications include:
- The details of how the PREMIUM framework enables the generation of personalized responses with LLMs.
- Advantages of tag-based personalization compared to alternative solutions.
- Practicality of preference feedback in real-world applications and its benefits over other feedback types.
- PREMIUM achieves higher data efficiency, better privacy security, and data accessibility compared to existing methods.
- Implementation details of the Personalized Movie Tagging experiments.
- Novelty of PREMIUM-Embed compared to existing approaches.
- A mitigation strategy for preventing extra tags from negatively affecting personalized generation.
- Reasons for selecting open-source LLMs as the backbone models for PREMIUM.
- Contributions of the Ranking-TAGER dataset.
- Definition and scope of "LLM personalization" in the context of our work.

We believe these enhancements, grounded in the reviewers' constructive feedback, significantly improve our submission. We respectfully request that the reviewers and area chair reconsider the overall contributions of our paper and the improvements made.

We hope these efforts demonstrate the value of our work to the research community, and we sincerely thank you for your thoughtful consideration.

---

### Meta-Review · Area_Chair_56ne · 2024-12-19

**Metareview:**

The paper proposes a framework called PREMIUM for achieving LLM personalization through individual-level preference feedback. PREMIUM utilizes a tag-based system to represent user profiles and preferences, guiding the LLM to generate more user-specific responses. The framework is designed to be lightweight and deployable on devices with limited computational resources, which is highlighted as one of its advantages. A key component of the work is a new dataset, Ranking-TAGER, intended to evaluate LLM personalization methods using a tag-based approach. Experimental results presented in the paper demonstrate that PREMIUM outperforms several baselines across various datasets in terms of accuracy, win rates, and adaptability to dynamic user preferences.

The paper’s primary strengths lie in its focus on an important and timely topic—LLM personalization—and its effort to create a lightweight, LLM-agnostic solution. The proposed framework, PREMIUM, demonstrates efficiency and scalability, which could have practical value in real-world applications where computational resources are limited. The introduction of Ranking-TAGER as a benchmark dataset for LLM personalization is also a potentially valuable contribution to the community, given the need for standardized evaluation protocols in this space.

However, there are significant weaknesses that prevent this paper from being ready for acceptance. First, several reviewers expressed concerns about the novelty of the work, particularly the reliance on embedding-based tag recommendation, which is well-studied in the recommendation systems domain. The reviewers found the use of tags for personalization to be somewhat limited in scope and applicability, as it imposes constraints on expressiveness and granularity. Additionally, the authors did not adequately justify the real-world feasibility of the proposed approach, given that the method requires users to provide preference feedback for approximately 30 rounds, which was considered burdensome and impractical.

The evaluation methodology also raised questions, particularly around the design of the Ranking-TAGER dataset. The use of AI annotators for generating feedback and the abstraction of dynamic user preferences through pre-defined tags were seen as oversimplifications that fail to capture the complexities of real-world scenarios. Furthermore, the experimental settings and baselines chosen for comparison were deemed insufficient by some reviewers, as they did not include stronger, proprietary LLMs or state-of-the-art soft-prompt tuning methods. Finally, while the paper highlights privacy and computational efficiency as advantages of the framework, these claims were not robustly supported by experimental evidence or concrete comparisons with alternative approaches.

In summary, while the paper addresses an important problem and introduces some interesting ideas, the weaknesses in novelty, practical applicability, and experimental rigor outweigh its contributions. Therefore, I recommend that the paper not be accepted at this time.

**Additional Comments On Reviewer Discussion:**

During the rebuttal period, the authors made significant efforts to address the reviewers’ concerns, providing detailed clarifications and conducting additional experiments. For example, they explained how the tag system enables personalization without requiring textual user profiles and argued that ranking feedback offers privacy and data efficiency advantages compared to alternative approaches. The authors also provided additional evaluations on other LLMs and conducted new experiments on recommendation tasks using the Amazon Review dataset. These efforts were commendable and addressed some minor concerns.

However, critical issues remained unresolved after the discussion phase. Reviewer 62bN maintained concerns about the practicality of requiring users to provide ranking feedback for many rounds, as well as the limited effectiveness of tag-based personalization. Reviewer j6HP was unconvinced by the claims of novelty, suggesting that the work relies heavily on existing embedding-based approaches and does not go far enough to advance the field of LLM personalization. Both reviewers also emphasized that the experimental setup did not adequately demonstrate the robustness of the proposed method in more complex, real-world scenarios. Reviewer CwzU raised issues with the oversimplified nature of the Ranking-TAGER dataset, arguing that it lacks realism and that its reliance on AI annotators undermines its value as a benchmark.

While the authors attempted to address these points in their rebuttal, the responses were often insufficiently persuasive. For instance, the claim that ranking feedback is readily accessible and unbiased did not alleviate the concern that it places a significant burden on users. Similarly, while additional experiments were conducted, they were not enough to fully address concerns about the evaluation methodology or the lack of comparisons with stronger baselines. Ultimately, the weaknesses raised by the reviewers were substantive and remained largely unresolved, which weighed heavily in my final decision to recommend rejection.

---

### Decision · Program_Chairs · 2025-01-22

Reject